# Aquifer configuration and geostructural links control the groundwater quality in thin-bedded carbonate/siliciclastic alternations of the Hainich CZE, Central Germany

Bernd Kohlhepp[1], Robert Lehmann[1], Paul Seeber[1], Kirsten Küsel[2,3], Susan E. Trumbore[4] and Kai U. Totsche[1]

[1]Hydrogeology, Institute of Geosciences, Friedrich Schiller University Jena, Burgweg 11, 07749 Jena, Germany
[2]Aquatic Geomicrobiology, Institute of Ecology, Friedrich Schiller University Jena, Dornburger Strasse 159, 07743 Jena, Germany
[3]German Centre for Integrative Biodiversity Research (iDiv), Halle-Jena-Leipzig, Deutscher Platz 5d, 04103 Leipzig, Germany
[4]Biogeochemical Processes, Max Planck Institute for Biogeochemistry Jena, Hans-Knöll-Str. 10, 07745 Jena, Germany

*Correspondence to:* Kai U. Totsche (kai.totsche@uni-jena.de)

**Abstract**
The quality of near-surface groundwater reservoirs is controlled, but also threatened, by a manifold of surface-subsurface
interactions. Vulnerability studies, typically evaluate the variable interplay of surface factors (land management, infiltration
patterns) and subsurface factors (hydrostratigraphy, flow properties) in a thorough way, but disregard resulting groundwater
quality. Conversely, hydrogeochemical case studies that address chemical evolution of groundwater often lack a
comprehensive analysis of the structural buildup. In this study, we aim to reconstruct the actual spatial groundwater quality
pattern from a synoptic analysis of the hydrostratigraphy, lithostratigraphy, pedology and land use in the Hainich Critical
Zone Exploratory (Hainich CZE). This CZE represents a widely distributed, yet scarcely described setting of thin-bedded
mixed carbonate-siliciclastic strata in hillslope terrains. At the eastern Hainich low-mountain hillslope, bedrock is mainly
formed by alternated marine sedimentary rocks of the Upper Muschelkalk (Middle Triassic) that partly host productive
groundwater resources. Spatial patterns of the groundwater quality patterns of a 5.4 km long well transect are derived by
principal component analysis and hierarchical cluster analysis. Aquifer stratigraphy and geostructural links were deduced
from lithological drill core analysis, mineralogical analysis, geophysical borehole logs and from mapping data. Maps of
preferential recharge zones and recharge potential were deduced from a digital (soil) mapping, soil survey data and field
measurements of soil hydraulic conductivities (Ks). By attributing spatially variable surface and subsurface conditions, we
were able to reconstruct groundwater quality clusters that reflect the type of land management in their preferential recharge
areas, aquifer hydraulic conditions and cross-formational exchange via caprock sinkholes or ascending flow. Generally, the
aquifer configuration (spatial arrangement of strata, valley incision/outcrops) and related geostructural links (enhanced
recharge areas, karst phenomena) control the role of surface factors (input quality and locations) versus subsurface factors
(water-rock interaction, cross-formational flow) for groundwater quality in the multilayered aquifer system. Our
investigation reveals general properties of alternating sequences in hillslope terrains that are prone to form multi-layered
aquifer systems. This synoptic analysis is fundamental and indispensable for a mechanistic understanding of ecological
functioning, sustainable resource management and protection.
**Keywords**
Critical zone, fractured rock aquifers, soil hydraulic conductivity, hydrostratigraphy, infiltration, intrastratal karst,
lithostratigraphy, Muschelkalk, vulnerability

42

**1 Introduction**

Near-surface groundwater reservoirs are increasingly threatened by anthropogenic impact, like the intensification of agricultural land use and water production, while their ecological functioning is still rather unexplored. Generally, the hydrogeochemical and (micro)biological compositions of shallow (< 200 m) groundwater systems is a result of biogeochemical processes and the fluid-rock/soil interactions while traveling through the overburden of the aquifers (Urich, 2002; Canora et al., 2008). Infiltrating precipitation, as the main component of recharge of surface near groundwater systems, first passes the soils. Thereby, the seepage collects among others, dissolved organic carbon (Guggenberger and Kaiser, 2003), colloidal and mineral, organic and organo-mineral suspended particles (Münch et al., 2002; Totsche et al., 2006, 2007; Schrumpf et al., 2013; Gleixner, 2013) and even biota (Dibbern et al., 2014).The residence time of the percolating water in the soils as well as in the unsaturated and saturated zones, regulates the chemical and biological quality by the extent to which the partial equilibrium of the rate-limited dissolution, retardation and release processes is achieved (Weigand and Totsche, 1998; Münch et al., 2002). Residence times and thus groundwater quality depend on various surface factors (duration and intensity of precipitation, land management/vegetation, presence of macropores) and subsurface factors (i.e. soil thickness, hydraulic parameters and preferential flow paths).

However, subsurface factors are spatially very variable in fractured rock and carbonate-rock terrains (Cook, 2003) and their relative importance for recharge and groundwater quality is still poorly understood (Gleeson et al., 2009). Sustainable management and resource protection therefore call for a holistic investigation of surface/subsurface architecture and interactions for the reconstruction of the hydrogeological functioning and groundwater quality. Concepts for vulnerability assessment (Hölting et al., 1995; Goldscheider et al., 2000; Witkowski et al., 2003; Vías et al., 2006), for instance, already consider the variable interplay of surface/subsurface factors and compartments in a thorough way, but disregard the evolution of pristine groundwater quality. Conversely, hydrogeochemical case studies (i.e. Suk and Lee, 1999; Moore et al., 2009) often lack a comprehensive analysis of the structural buildup and composition of the entire compartments, whereas the important "biogeoreactor" of the overburden soils are vastly neglected. Studies that include land use and soil groups (Pronk et al., 2009; Steward et al., 2011; Allocca et al., 2014), typically leave groundwater chemistry out of focus. In this study, we follow a holistic approach for reconstructing the pristine groundwater quality in the aquifers of the Hainich Critical Zone Exploratory (Hainich CZE) in central Germany. Here, the bedrock of fractured, mixed carbonate-/siliciclastic alternations represents a widely distributed, yet scarcely described geological setting. We synoptically characterize all surface and subsurface compartments involved in the flow of water and transport of matter in the subsurface of the Hainich CZE. The goals of our study are to (1) understand the hydrogeological and biogeochemical functioning of the subsurface in a so far not adequately addressed setting provided by the Hainich CZE, (2) explore the links and feedbacks between surface and subsurface, and (3) to demonstrate that a holistic multi-method-approach, that considers the surface and subsurface factors, is indispensable for the mechanistic understanding of the coupling of surface and subsurface compartments, fluid dynamics, biogeochemical element cycling and ecology in the critical zone.

To reach these goals, the following research questions are answered: (I) How is the critical zone comprised and connected in the hillslope setting of a thin-bedded, mixed carbonate/siliciclastic succession? (II) How do spatial arrangement, particularly outcrop patterns and geostructural links (karst features like caprock sinkhole lineaments) impact compartment connection, (intrinsic vulnerability) and groundwater quality? Do groundwater quality in contrasting summit/midslope and footslope wells reflects surface influences?" And finally: (III) what are the main control parameters for groundwater quality? What are the reasons for particular hydrogeochemical conditions within the multi-storey/hillslope aquifer system?

**2 Site description**

The Hainich Critical Zone Exploratory covers 430 km² of a hillslope subcatchment of the Unstrut river in northwestern

Thuringia, Central Germany (Fig. 1). It is bounded by the recipient Unstrut river and the distribution area of the Upper
Muschelkalk formations. The Hainich represents a NW-SE orientated geological anticline that is developed topographically
as a low mountain range with a steep western, and a moderately inclined eastern flank (Jordan and Weder, 1995). The study
area is located at the eastern flank of the Hainich hillslope, that shows a geostructural buildup with NE dipping strata in the
direction of the syncline of Mühlhausen-Bad Langensalza (Kaiser, 1905; König, 1930; Patzelt, 1998; Wätzel, 2007). A
tectonical uplift, faulting and tilting of strata is assumed for the Late Cretaceous in analogy to the surrounding horst
structures Thüringer Wald (in the S) and Harz (in the N; compare to Voigt et al., 2004; Kley and Voigt, 2008). The
outcropping strata in the study area comprise sedimentary rocks (Middle/Upper Muschelkalk and Lower Keuper - subgroups
of the Middle Triassic). According to the German stratigraphy (Deutsche Stratigraphische Kommission, 2002), the Diemel
formation (Middle Muschelkalk), Trochitenkalk, Meissner, and Warburg formation (Upper Muschelkalk) and the Erfurt
formation (Lower Keuper) outcrop in the study area. The Upper Muschelkalk subgroup, which hosts the target aquifers of
the Hainich CZE, is organized by bio- and lithostratigraphic marker beds (Ockert and Rein, 2000; Kostic and Aigner, 2004).
Previous studies hydrostratigraphically organize the Upper Muschelkalk subgroups into a Hainich Transect Lower Aquifer
Assemblage (HTL) and a Hainich Transect Upper Aquifer Assemblage (HTU; Küsel et al., 2016).The area belongs to the
Cfb climate region (C: warm temperate, f: fully humid, b: warm summer) according to the Köppen-Geiger classification
(Kottek et al., 2006) and exhibits a leeward decline in areal precipitation and increasing mean air temperature from the
Hainich ridge (> 900 mm/y; 7.5-8 °C) to the Unstrut valley (< 600 mm/y; 9-9.5 °C; long term average 1970-2010, TLUG,
2016). The intensively investigated study area is limited to a 29 km$^2$-subarea of the Hainich CZE, that surrounds the soil and
groundwater monitoring transect (Küsel et al., 2016).
The Hainich CZE is a multifarious environmental laboratory for multiscale geo- and bioscience research, as
(1) both, the so scarcely described geological setting (alternations of marine thin-bedded limestones and marlstones) and the
hillslope relief/sloping aquifer configuration are common and widely distributed
(2) the monitoring plot and well transect (see Küsel et al., 2016) provides an unique access to the multi-layered aquifer
system of the common setting
(3) it represents a rare anthropogenically low-impacted (non-contaminated) cultural region in central Europe with a very
extensive type of land management during the last centuries, allowing the investigation of natural (surface) signal
transformation in the pristine aquifer systems and of ecosystem functioning
(4) the geostructural and lithological properties were found to be predictable and thus enable the tracking of
biogeochemically cycling and quality development within single aquifer storeys
(5) the Hainich it is a regionally important groundwater recharge area in Thuringia (Fig. 1) and an example of peripheral
groundwater supply for water deficient sedimentary basins (Rau and Unger, 1997; Hiekel, 2004)
**3 Material and methods**
For the detailed and synoptic analysis of the critical zone and the aimed reconstruction of the groundwater quality control
factors, we apply a multi-method approach that comprises drill core and geophysical-log analysis, geological surveys, soil
and groundwater samplings, soil hydraulic measurements and a statistical analysis of groundwater hydrogeochemistry.
**3.1 Groundwater well transect**
The Upper Muschelkalk strata are accessed by a hillslope transect that consists of multiple wells at five sites for groundwater
monitoring, termed H1 to H5. Each site contains 1 to 4 drilled wells. Site H1 (wells H11 to 14) and H2 (H21 to H23) are
located in the summit/shoulder region of the catchment, which is covered by forest. Sites H3 (H31/32), H4 (H41 to H43 and
additional sweep well H4SB) and H5 (H51 to H53) are situated in the agriculturally used, midslope and footslope regions of
the Hainich low mountain range.
**3.2 Hydrogeochemistry and hydraulics**
Groundwater levels were recorded by permanently installed data loggers (Orpheus Mini, Ecolog 500/800, OTT Hydromet
GmbH, Germany). Groundwater was sampled at least every four weeks over three hydrological years (Nov. 2013 to Oct.
2015) in 15 permanent monitoring wells (5.1 to 88.5 m final depth below surface) of five sites along the 5.4 km long well
transect (Fig. 1) between 417 and 244 meters above mean sea level (masl). The groundwater wells were sampled with
submersible motor pumps (MP1, Grundfos, Denmark) or, in the case of very low water levels with bladder pumps/bailers.
The physicochemical parameters of groundwater samples were measured on site using a flow-through cell, equipped with
probes for temperature (T), pH (SenTix 980, WTW GmbH, Germany), electrical conductivity, temperature corrected to 25°C
(EC_25)(TetraCon 925), redox potential (Eh) (SenTix ORP 900), and dissolved oxygen ($O_2$) (FDO 925). Hydrochemical
analyses (duplicates) comprised major and minor ions (< 0.45 μm, PES filter) by ICP-OES (725 ES, Varian/Agilent, USA)
and ICP-MS (Thermo Fisher Scientific, Germany; Thermo Electron, U.K.), acid and base neutralizing capacity by acid/base-
titration, major anions ($SO_4^{2-}$,$Cl^-$, $NO_3^-$, $NO_2^-$, $PO_4^{3-}$; PES filter < 0.45 μm) by ion chromatography (DX-120, DIONEX,
USA), redox sensitive parameters ($Fe^{2+}$, $NO_2^-$, $NH_4^+$, $HS^-$) by colorimetry (DR/890, Hach, USA) and determination of carbon
sum parameters (TOC, TIC, DIC, DOC; PES filter < 0.45 μm) by high temperature catalytic oxidation (multi 18 N/C 2100S,
AnalytikJena, Germany). Groundwater was classified using a Piper plot (Piper, 1944) according to Furtak and Langguth
(1967) and Kralik et al. (2005).
**3.3 Statistical analysis**
SPSS 22 (IBM Corp., USA) and Origin Pro 2015 (OriginLab Corp., USA) were used for descriptive and multivariate
statistics of the hydrochemical data. This includes hierarchical cluster analysis (HC) for distinguishing hydrogeochemical
groups independently from lithostratigraphy-based hydrostratigraphy. Two different parameter sets were statistically
examined at the same number of groundwater analysis from three hydrological years. A complete parameter set (a),
including all measured parameters, was chosen to identify theparameters that control the hydrochemical compositions of the
groundwater domains by means of Principal Component Analysis (PCA). As redox processes seems to be very distinct in the
studied groundwaters (masking other processes), a limited parameter set (b) was defined that does not include ions which are
strongly affected by redox processes. The complete parameter set (a) contains, EC_25, pH, $O_2$, Eh, TOC, anions ($NO_3^-$, $Cl^-$,
$SO_4^{2-}$), cations ($Ca^{2+}$, $Mg^{2+}$, $Mn^{2\ and\ 4+}$, $K^+$, $Na^+$, $Fe^{2\ and\ 3+}$, $Zn^{2+}$, $Sr^{2+}$, $Ba^{2+}$) and silica ($Si^{4+}$). The limited parameter set (b)
includes EC_25, pH, $Cl^-$, $Ca^{2+}$, $Mg^{2+}$, $K^+$, $Na^+$, $Zn^{2+}$, $Sr^{2+}$, $Ba^{2+}$ and $Si^{4+}$. Data preparation for HC includes a z-score
normalization of concentrations. Strongly correlating variables (> 0.9) and constant values were excluded, outliers were
eliminated (next neighbor method) and listwise deletion was carried out (according to Backhaus et al., 2016). Then, a
euclidic distance measure was applied and five clusters were chosen based on the visual inspection of the scree plot elbow
criterion (Cattell, 1966). The phenon line was chosen at a linkage distance of about 67. Thus, samples with a linkage distance
lower than 67 were grouped into the same cluster. Finally, Ward´s method for clustering was applied (Ward, 1963).
**3.4 Field survey**
482 bedrock outcrops were investigated for lithology, flow paths, karst features and the thickness of quarternary cover
sediment. Strike and dip directions were constructed, based on the mapped boundary between two prominent formations
(Trochitenkalk/Meissner formation). Soil profiles were described for 117 ramcore drillings (GSH 27, Robert Bosch GmbH,
Germany; 50 and 60 mm diameter, drilled to the bedrock; up to 5 m depths), and 154 supplementary soundings (Pürckhauer,
22 mm). Soil description was carried out according to the German soil survey instruction (Bodenkundliche Kartieranleitung
KA5, AG Boden, 2005) and the world reference base (WRB)-scheme (IUSS Working Group WRB, 2006). Soil colors were
determined using a Munsell soil color chart.
Soil hydraulic conductivities ($K_s$) were measured in two depths (10 and 30 cm) at 16 locations (duplicates) within the
potential groundwater catchment area by using a Guelph-Permeameter (2800K1, Soilmoisture, USA). Soil hydraulic
conductivities were calculated according to the equations in Elrick and Reynolds (1992, p. 320 ff.). Hydraulic conductivities
for each site are calculated as the harmonic average of the topsoil and subsoil (or parent rock in Ah/C-soils). Mean values of
hydraulic conductivities of soil groups are given as spatially weighted arithmetical means.
**3.5 Analysis of drill cores and borehole logs**
For the determination of the stratigraphic succession, aquifer properties and mineralogical indicators for groundwater flow,
395 meters of drill cores from twelve drillings were investigated. Lithology, fractures/pores and rock weathering were
described according to DIN EN ISO 14689-1. We extended this geotechnical rock classification to sedimentary structures,
limestone classification (Dunham, 1962), pore classification (Lucia, 1983), degrees of karstification, aquifer types, fracture
angles and fracture colors, as well as secondary mineralization on fractures. Additional information about stratigraphy, clay
content, fracturing and groundwater inflow was revealed by analyzing geophysical open-borehole logs of ten drillings (site
H1/2/3/4/5, Fig. 1). This included data on caliper, passive gamma-ray radiation, sonic velocity (delay time of sound waves),
specific electrical resistivity of rocks, as well as the temperature and specific electrical resistivity of the well water. Gamma-
ray curves are interpreted, based on the graphical interpretation of high gamma ray peaks (marlstones) and intersections of
gamma ray curves with an empirically defined "shale line", here at 90 API (separating limestones from marlstones).
Rock-forming minerals were analyzed by X-Ray diffraction (Bruker D8 Advance,Cu-Kα, 40 kV, 40 mA, Bruker AXS Inc.,
USA) and Fourier transform infrared spectroscopy (FTIR; Nicolet iS10; Thermo Fisher Scientific; USA).
**3.6 Grain size analysis of soils**
Grain size analysis of all major soil groups (144 duplicates) were carried out on filtered (< 125 μm), decarbonized (HCl) and
organic-free ($H_2O_2$) samples (Laser Particle Sizer, Analysette 22, Fritsch, Germany).
**3.7 Geospatial analysis and construction of preferential groundwater recharge areas**
Maps of soil groups and geology, were created and jointly analyzed with aerial imagery and a digital elevation model (DEM,
2 m resolution) using ArcGIS 10.3 (ESRI Inc., USA). The primary data sets were interpreted in terms of land use types,
surface morphology and drainage patterns with special focus on karst phenomena like caprock sinkholes, which are
considered as preferential input structures for funneled infiltration (Nennstiel, 1933; Mempel, 1939; Hecht, 2003). The two-
dimensional correlation of geophysical and geological well logs was carried out with well management software GeoDIN
V.8 (Fugro Consult GmbH, Germany) by graphical correlation of prominent high gamma-ray peaks and stratigraphic marker
beds (grainstones/rudstones). The spatial correlation of marlstones presumes that basin-center marlstones are laterally more
continuous than shallow water limestones (Aigner, 1985). According to published examples for multi-storey subsurface
architecture (Haag and Kaupenjohann, 2001; Heinz and Aigner, 2003; Klimchouk, 2005; Sharp, 2007), we use the term
"aquifer storey" to emphasize our conceptualization of the fine-stratified setting by arbitrary definition of intervals that are
dominated by fractured limestone beds and confined at the top and base by unfractured or low permeable beds, the latter
with an effective minimum thickness of 80 cm. The degree of aggregation is a compromise between increased detailedness
to account for different hydrogeochemical patterns and the necessary well yield for recovering water samples.

**3.8 Soil group map**

The soil group map was constructed from different geospatial data sources including (1) land management map (interpreted from satellite images and field observations), (2) outcrop areas of surface geology (mapping data) and the (3) DEM2. After conversion into raster data, a (4) slope gradient map and a (5) topographical position index (TPI; Weiss, 2001) map with an empirically chosen radius parameter of 75 meters were calculated using the "Slope"-function of the ArcMap toolbox. The slope position/relief position (i.e. shoulder, midslope) was calculated using the "Relief Analysis"-tool of ArcGIS (Deumlich et al., 2010). It considers slope gradient, relative elevation and profile curvature (concave/convex) by using the slope gradient (4) and the TPI-raster (5). The spatial distribution of soil groups was calculated with a query function, which considers land management, slope position and slope gradient for the following substrata: "carbonate soil series", "mixed carbonate-siliciclastic soil series", "claystone series" and "loess loam series".

**3.9 Groundwater recharge potential map**

For the determination of preferential zones for infiltration/recharge, we calculated qualitative maps of the recharge potential (compare to Muir and Johnson 1979; Shaban et al., 2006; Deepa et al., 2016) in ArcGIS 10.3 by a weighted linear combination of surface/subsurface properties that influence water infiltration and percolation. The input raster datasets and the respective weight factors were chosen based on expert knowledge and the best fit to measured soil hydraulic conductivities: aquifer storey overburden thickness (40 %), type of bedrock (limestone-dominated vs. marlstone-dominated strata; 15 %), soil classes (15 %), fracture/karst zones (10 %); vegetation and type of land management (10 %), soil thickness (5 %) and slope angle classes (5 %).

**4 Results**

**4.1 Surface characteristics of Hainich CZE**

**4.1.1 Relief and surface water network in the groundwater catchment**

The study area ranges from the Hainich ridge (topographic water divide) towards the Unstrut valley and covers altitude regions between 170 and 494 masl with an average slope of 35 m/km. Two orders of relief types are here distinguished: the first order (regional) relief grades from the culmination of the Hainich low mountain range towards the low-angle midslope (<5°) and footslope. The second order (local) relief is formed by a bundle of ten straight, parallel and roughly equidistant, SW-NE oriented transverse valleys with slopes >5°. Second order relief elements are also the two NW-SE oriented lineaments of more than eighty caprock sinkholes (which are passive karst phenomena) with up to 80 m in diameter. Caprock sinkholes are mostly exhibited on local ridges. A second and parallel lineament of three shallow elongated (uvala-like) karst depressions with a horizontal extent of up to 400 m crosses the lower Hainich hillslope (Fig. 2 A). A few small contact springs occur in the study area. Larger springs occur at the lower eastern slopes of the Hainich ridge, tracing regional fault/fracture zones. Headwater areas of creeks, small rivers, transverse valleys and even agricultural drainage ditches are mostly dry. During our monitoring period (November 2013 to May 2016) stream runoff typically occurred from December to March.

**4.1.2 Land management and history of land management**

Major types of land management are agriculture (crop, pasture) and forest (unmanaged and managed deciduous forest). Forests are to a large extent within the unmanaged Hainich National Park that occupies summit to midslope positions. Extensive forest management has been the dominant type of land management during all time periods even with no deforestation during the medieval ages (Otto, 2000). Mode of forest operation shifted from random selection of wood via coppice use (since the 15[th] century) and planter forestry (since the 20[th] century) to unmanaged woodland with the foundation

of the Hainich National Park in December 1997 (Otto, 2000; Röhling and Safar, 2004). Parts of the study area (12 %), which
are actually unmanaged grassland/scrubland areas within the Hainich National Park, had been formerly used as a military
training area since 1964, particularly for tank trainings from 1980 until 1990 (Otto, 2000; Poser, 2004). Cultivated grasslands
outside the Hainich National Park, which are used both as meadows and pastures, cover parts of the midslope and locally
some cleared glades at the shoulder as well as riparian areas of small rivers. Cropland (locally used for wheat, corn and
canola production) covers mainly the midslopes and footslopes (Fig. 2 A). Agricultural plots are typically large due to
organization by GDR´s agricultural production cooperatives (from 1945 to 1990). These plots are now (since 1990) managed
by privately owned agricultural cooperatives.
**4.2 Soils in the groundwater catchment**
**4.2.1 Soil distribution and soil development**
Soils cover the entire landscape with major soil series developed from carbonate rocks ("carbonate soil series": Rendzic
Leptosols to Chromic Cambisols), siliciclastic rocks ("siliciclastic soil series": Luvisols, Stagnosols) or alluvial sediments
(WRB and German soil groups: Table 1). Culmination areas and adjacent shoulder positions are covered with Cambisols.
Small plateaus and spurs between transverse valleys in shoulder positions exhibit Chromic Cambisols which grade into
Cambic Regosols on the shoulder and Calcaric Regosols on the midslope. Chromic Cambisols are also found in local
depressions and old caprock sinkholes. Rendzic Leptosols occur in form of narrow patches in western crestal areas (close to
well H11). Luvisols are coupled to the spatial distribution of loess loam in the central and eastern midslope/footslope areas.
In case of a very thin loess loam cover, soils are developed as Pelosol-Cambisol and Cambisol. Fluvial soils cover the central
parts of headwater areas and the complete valley floor in the lower parts (in the northeast) of the study area. Colluvisols
occur at the margins of local valley flanks in the shoulder and midslope area (Fig. 2 B). Typical sequences of two
superimposed soils comprise (I) Chromic Cambisols (paleosols) developed from marlstones and (II) Luvisols developed
from loess loam with windblown loess sedimentation after the formation of soil (I) (Fier, 2012). Average soil thicknesses are
21 cm (for Rendzic Leptosols), 32 cm (Rendzic Leptosol-Cambisol transitions), 66 cm (Cambisols), 54 cm (Chromic
Cambisols), 71 cm (Luvisols), 132 cm (Fluvisols), 91 cm (Stagnosols) and 78 cm (Colluvisols). Subsoils show higher
average clay and fine silt content compared to the topsoils of the same soil group (Fig. 3).
**4.2.2 Soil hydraulic properties**
Average (median) soil hydraulic conductivities (Ks) of the five major soil groups infer, that Cambisols and Luvisols (1.3 to
$1.5* 10^{-4}$ m/s) form the most conductive soil cover in the study area, followed by Rendzic Leptsols and Chromic Cambisols
(2.5 to $5* 10^{-5}$ m/s) and Stagnosols (about $6* 10^{-7}$ m/s; Table 1, Fig. 3 and 8). Topsoils of Chromic Cambisols are
considerably more conductive than those in Rendzic Leptosols and Luvisols. Generally subsoils are less conductive than
topsoils of the same location. Soil hydraulic conductivities (Ks) are essentially uncorrelated with the soil texture (i.e.
correlation Ks vs. Median grain size: Spearman $r^2 = + 0.17$). Soil thickness is uncorrelated to the slope gradient ($r^2 = -0,24$
and barely better if slope positions are correlated individually).
**4.2.3 Geological outcrop zones**
The landscape of the Hainch low mountain range has been developed within the Triassic formations of the Upper
Muschelkalk and Lower Keuper. All formations outcrop in distinct zones along the eastern slope with dip angles steeper than
the slope: from 5 ° (footslope) to 8 ° (shoulder position); and lower/older strata outcrop in higher relief positions (Fig. 2 A).
A NW-SE orientated normal fault is located close to the Hainich summit with offsets of about 10 m (Fig. 2A). Fault-bound
troughs (up to 700 m long and 150 m wide) are also found parallel to this fault. The Triassic rocks are covered by young
loess loam in sheltered, concave depressions of the east-exposed slopes. Alluvial soils and colluvia fill the valley bottoms
with increasing thickness (maximum 3.5 m) and areal cover from the midslope to the footslope (Fig. 2A).

## 4.3 Subsurface properties of Hainich CZE

### 4.3.1 Lithostratigraphy and bedrock mineralogy

The stratigraphic succession is characterized by thin-bedded carbonate-siliciclastic alternations of which only the limestone
beds are fractured and therefore represent potential aquifers. The Diemel formation (mmDO) comprises thin (1-3 cm)
yellowish dolomitic marlstones, fine crystalline dolomite and rarely cavernous dolomites without any fossils, gypsum or salt.
The 7 m thick Trochitenkalk formation (moTK) with thick (5-30 cm), gray, coarse bioclastic limestones (mainly rudstones
with the rock-forming fossil *Encrinus liliformis*) forms a carbonate-rock-fracture aquifer with minor karstification
(intrastratal karst according to Ford and Williams, 2007). Large scale (meter-scale) and consistent fractures, a high fracture
index and dissolution-enlarged fractures and vugs (cf Lucia, 1983), as well as karst breccia at the base of the formation are
common features. The fractures found in rock cores are mainly developed as stratabound (Odling et al., 1999) fractures that
are restricted to the limestone beds, as common in limestone-marlstone alternations (Meier et al., 2015). All fractures and
pores are stained with red brownish (10R6/6) coatings. Karst features in the Trochitenkalk formation occur from the near
surface down to 90 m depth. Rock mineralogy is dominated by calcite and very low contents of dolomite with trace amounts
of quartz, muscovite, chlorite and Na/K-feldspar. Based on a consistent (moTK) formation thickness, bed thickness and
limestone types (cf Dunham, 1962) in all wells, lateral facies changes within the Trochitenkalk formation are negligible.
The 34.6 m thick Meissner formation (moM) consists of 5 m thick basal marlstones with three discrete bioclastic limestone
beds (Kalkbank α/β/γ), overlain by 29 m of alternating thin (2-5 cm) limestones (mudstones to grainstones) and marlstones.
These are covered by a thick (> 60 cm) bioclastic, regional biostratigraphic marker horizon, the Cycloidesbank, which is a
known regional aquifer (Grumbt et al. 1997; Hecht, 2003) of minor importance. Limestones in the Meissner formation are
fracture aquifers and their rock matrices are predominantly composed of calcite and trace amounts of dolomite, quartz, illite
and feldspar. The overlying 16 m thick Warburg formation (moW) with predominantly marlstones (mineralogically: calcite,
dolomite, quartz) and the 35 m thick Erfurt formation (kuE) with dolomite rocks (dolomite, calcite and quartz), siltstones,
silty sandstones and claystones (illite, muscovite, chlorite and gypsum), represent both unfractured aquitards (Hoppe, 1952;
Hecht, 2003) and form the low permeable top-seal strata of the Upper Muschelkalk aquifer storeys at the footslope positions
(Table 2). The Triassic strata are bounded by an erosional unconformity on top, which is overlain by aeolian loess loam
developed from loess deposits of the last glacial period (Weichsel glacial in Germany; Greitzke and Fiedler, 1996) and
alluvial/colluvial sediments of Holocene age (Rau and Unger, 1997).

### 4.3.2 Hydrostratigraphy

The correlation of sedimentological core logs and geophysical borehole logs infers that a multilayer hydrostratigraphy is
applicable and all aquifers and aquitards are presumably continuous (and with similar thickness and limestone type) on the
scale of the research transect (Fig. 4). In the Upper Muschelkalk strata (Trochitenkalk formation (moTK) and Meissner
formation (moM)), the alternated bedding of limestones, marlstones and claystones result in permeable fracture aquifers
(dense matrix, varying fracturing) and low permeable marlstones (dense matrix, unfractured). Ten aquifer storeys are newly
defined for the Hainich CZE. Of these, the Trochitenkalk formation contains one aquifer storey (moTK-1) and the Meissner
formation contains nine aquifer storeys (moM-1 to moM-9; Table 2). All footslope aquifer storeys are overlain by
impermeable cap rocks of the Warburg formation (Upper Muschelkalk) and the Erfurt formation (Lower Keuper). Flow
paths are predominantly fractures and matrix porosity is lower than 5 % in all stratigraphic intervals. Although it is of karst-
fracture type with partially solution-enlarged fractures, the karstification and the development of conduits is limited and
concentrated at the formation´s very base. Below moTK-1, the impermeable dolomitic marlstones of the Diemel Formation
(mmDO) form a hydraulic basis seal. We observed an intact and unweathered base seal in all wells that were drilled to the
base of the moTK-1. The Meissner formation (moM) contains limestone-fracture aquifers which are interbedded marlstone-
aquitards on the decimeter to meter scale. Limestones of this formation are almost exclusively fracture aquifers with very
little matrix porosity, concentrated at certain thickly bedded limestone marker beds.
**4.3.3 Land use types and soils within the outcrop zones of aquifer storeys**
The outcrop zones of the two lowermost aquifer storeys (moTK-1, moM-1) are predominantly covered by forest, whereas
the outcrop zones of stratigraphically higher storeys show mixed types of land management including, forest, cropland and
pasture (Fig. 5 A) Within the aquifer outcrop zones, the four major soil groups (Rendzic Leptosol, Leptic Cambisol,
Cambisol and Chromic Cambisol) are present in different spatial proportions (Fig. 5 B). Significant proportions of Luvisol
cover the outcrop areas of moM-7, moM-9 and moTK-1. Stagnosol and Colluvisol are restricted to the outcrop area of mm,
moTK-1 and moM1-4. The grain size classes of subsoils show a slight trend of increasing silt content towards the outcrop
zones of stratigraphically higher aquifer storeys (moTK-1 to moM-9). Argillaceous soils occur frequently in moM-4 to
moM-6. Pure clay as subsoil category is restricted to moM-2 to moM-7 (Fig. 5 C).
**4.4 Groundwater chemistry**
**4.4.1 Groundwater classification**
The groundwater in the carbonate rock landscape is classified as little to strongly mineralized earthalkaline, bicarbonatic
type (site H1/2/3/4), and earthalkaline, bicarbonatic-sulfatic type (cf TLUG, 1996) for groundwater of the deep well H51
(moTK-1). The order of abundance for dissolved ions is: $Ca^{2+} > Mg^{2+} > Na^+ > K^+$ and $HCO_3^- > SO_4^{2-} > Cl^-$. According to the
groundwater subtypes of Kralik et al. (2005) the moM-waters (except for well H14) plot in the "dolomite"-field, moTK-1 -
waters (wells H13/21) and water from H14 (moM) plot in the "calcite"-field. All other moTK-1 wells (H31/41/51) are
classified as the mixed subtype ("calcite+dolomite"). Two contact springs in the recharge area (Grauröder Quelle,
Ihlefeldquelle) and two karst springs in the discharge area (Kainspring, Melchiorbrunnen, coupled to NW-SE orientated fault
zones, are classified as carbonate-earth alkaline type, predominantly in contact with calcite in the recharge area and a calcite-
dolomite mixing type in the discharge area. Groundwater chemistry in aquifer storeys in deeper stratigraphic positions,
exhibit high $O_2$, $SO_4^{2-}$, $Sr^{2+}$ and low $Mg^{2+}$, $HCO_3^-$, $Si^{4+}$, $K^+$, $Na^+$ and $NO_3^-$ concentrations. Shallow aquifers storeys are partly
anoxic in midslope/footslope positions and deeper aquifer storeys are oxic in all slope positions (Fig. 6). Aquifers in
footslope wells show higher concentrations of $K^+$, $Mg^{2+}$, $Si^{4+}$, $Sr^{2+}$, $SO_4^{2-}$ and $Cl^-$ compared to the same aquifer in summit
positions.
**4.4.2 Statistics of hydrogeochemistry and chemical composition of groundwater groups**
Hierarchical cluster analysis revealed 5 clusters. Cluster1 comprises the groundwater samples from upper slope moTK-1
wells (H13/21) and the moM-1 well H14. Cluster 2 encompasses the groundwater samples from moTK-1 wells H31/41 and
also from moM-5/6 well H32. Groundwater samples from moTK-1 well H51 plot in cluster 3. Cluster 4 includes moM-6/7/8
wells H42/43/4SB, while cluster 5 consists of two moM-3/4/8 wells of the same site (H52/53; Fig. 7). Multi-case clustering
does not occur. As a result of the principal component analysis (PCA) using the complete parameter set (Fig. 7 a), and the
parameter set without redox sensitive parameters (Fig. 7 b), data points fall into very similar groupings compared to the HC.
According to the PCA of the complete parameter set, the first two components (PC1 plus PC2) explain 54.9 % (variances of
PC1: 34.6 % and PC2: 20.4 %), and according to the parameter set without redox sensitive parameters PC1 plus PC2 explain
62.1 % (variances of PC1: 35.4 % and PC2: 26.8 %) of the total variability, respectively. For both PCA, the parameters with
the highest factor loads are $Si^{4+}$, $K^+$, $Na^+$, $HCO_3^-$, $Mg^{2+}$ for PC1 and $Sr^{2+}$, $SO_4^{2-}$, EC_25 and $Cl^-$ for PC2.
The average chemical composition of the five groundwater clusters is significantly different with respect to their physico-
chemistry and major ion composition. Cluster 1 shows high pH and Eh, as well as high concentrations in dissolved $O_2$, TOC
and $Ca^{2+}$. High concentrations in $HCO_3^-$, $NO_3^{2-}$. TOC, $Cl^-$ and $Mg^{2+}$ are distinctive for Cluster 2, whereas Cluster 3 is
characterized by very high $SO_4^{2-}$, $Ca^{2+}$, $Cl^-$, and $Sr^{2+}$ concentrations in combination with high redox potential and moderate
$K^+$ and $Na^+$ content. Cluster 4 shows very high concentrations in $Fe^{2+/3+}$, $Mn^{2+/4+}$, TOC, $Cl^-$ in combination with very low
redox potential and very low dissolved oxygen. The latter is also applied to Cluster 5, that additionally contains high
concentrations in $K^+$, $Na^+$ and $Mg^{2+}$ (Fig. 8).
**4.4.3 Fluctuations of groundwater levels and quality**
Groundwater levels are confined in footslope wells. In the wells of site H5, groundwater rises 30 m (H53), 50 m (H52) and
70 m (H51) higher than the base of the screen section. Fluctuations of groundwater levels in our monitoring wells range from
1-3 m in the hilltop recharge area (well site H13) to more than 25 m in the groundwater transit area (well site H5). The
groundwater level fluctuations show a strong seasonality with annual highstands (March to April) and lowstands (October to
December). Monitoring wells of different sites and screen depths differ in average concentrations of the major solutes. These
spatial differences are generally higher than the seasonal fluctuations in the wells. An exception to these conditions are
marked seasonal fluctuations of $Ca^{2+}$ (monitoring well H31/41), $Cl^-$ (H52/53), $K^+$ (H41), $Mg^{2+}$ (H41), $Na^+$ (H31/41), $Si^{4+}$
(H41) and $SO_4^{2-}$ (H31/41).
**4.4.4 Recharge potential map**
The recharge potential is here defined as a qualitative measure of the probability for infiltration, percolation and groundwater
recharge.The maps (Fig. 8) visualize potential spatial variation in infiltration-recharge, waiving spatiotemporal variable
conditions like precipitation characteristics and antecedent soil moisture. The recharge area maps of two selected aquifer
storeys (Fig. 8A and 8B) show the largest recharge potential for their outcrop areas on the Hainich hillslope. Increasing
overburden thickness is accompanied with a drastic decrease in recharge potential towards the NE. With respect to the relief
position, greatest recharge potential is assumed for local valleys and in the extension of sinkhole lineaments, although these
areas show increased soil thicknesses. In areas with greater thickness of the overburden strata, valleys are considered as areas
of preferential recharge. A slightly higher recharge potential is assumed for pasture and cropland areas compared to forests.
The same holds for the flat summit/culmination and upper slope areas while the slopes of transverse valleys show greater
surface water runoff.
**5 Discussion**
**5.1 Hydrostratigraphy**
**5.1.1 Aquifer structure evaluated by lithology**
The lithologically defined aquifer-aquitard succession in combination with the increase in karstification towards the base,
and the preferential presence of groundwater at the base of limestone packages point to confined aquifers with stratigraphic
flow control (Klimchouk and Ford, 2000; Goldscheider and Drew, 2007). As hydraulic heads and chemical groundwater
compositions of different aquifers at the same sites [H13/14 (cluster 1); H41/42 (cluster 2 and 4); H51/53 (cluster 3 and 5)]
are significantly distinct, vertical communication between aquifer storeys is highly reduced. Thus we interpret the thin
aquifer beds as confined, "sandwich flow type" aquifers (term: White, 1969; Klimchouk, 2005) which are interbedded with
confining aquitards. The assumed lateral continuity of aquifers and aquitards results in a "layer-cake" aquifer architecture
which has been demonstrated for the Upper Muschelkalk of Central Europe (Aigner, 1982, 1985; Merz, 1987; Simon, 1997;

Borkhataria et al., 2005). Secondary mineralizations, that indicate groundwater flow (for instance Liesegang banding), occur on limestone fracture walls. Fracturing of aquifer rocks increases close to the (NE-SW-striking) fault zones (Hoppe, 1962) and karstification follows the network of faults (Goldscheider and Drew, 2007). Red and brown iron and manganese oxides on upper to midslope fractures (H1/2/3; cluster 1 and 2) indicate temporarily unsaturated conditions, whereas green/gray fracture minerals in the footslope domains (H4/H5, cluster 4/5) point to permanently water-saturated conditions due the lack of oxidized Fe/Mn-minerals. Corrosion occurs in the form of intrastratal karstification, i.e. one or more layers of soluble strata is covered or sandwiched between insoluble beds (Palmer, 1995). Karstification is consistently high in the thick bedded aquifer storey (moTK-1) and less pronounced in thin-bedded aquifer storeys (moM-1/4/5/6; Fig. 4) as the bed thickness controls continuity, spacing and width of joints (Goldscheider & Drew, 2007). In general, the lithological definition of hydrostratigraphy allows a structuring with a high spatial resolution.

**5.1.2 Evaluation of groundwater chemistry**

Stratigraphically deep aquifers in summit positions are grouped in Cluster 1 (aquifer storey moTK-1 and moM-1). Cluster 2 (moTK-1 and moM-5+6) encompasses deep and stratigraphically intermediate aquifer storeys in midslope positions and cluster 3 (moTK-1) a deep aquifer storey in the footslope position. Shallow aquifer storeys (moM-6/7/8) in midslope position within a local valley are grouped in cluster 4 and intermediately deep aquifer storeys (moM-3/4/8) are grouped in cluster 5. According to this grouping, the slope position and the depth below surface is important, likewise to the depth level within the hydrostratigraphy (= the aquifer storey).

Principal component analysis (PCA) using the complete parameter set (a) infers, that each aquifer storey contains more than one type of groundwater chemistry (Fig. 7). By contrast, the PCA (b) carried out with the limited parameter set (without redox-related parameters) shows a clear separation of moTK-1 and moM-1-9 aquifer storeys (except cluster 2). According to the PCA, cluster 1 encompasses high factor charges in pH, Eh, $O_2$, $NO_3^-$ and low factor charges in $Mg^{2+}$, $Na^+$, $K^+$, $Si^{4+}$. The same is less markedly applied for cluster 2 whereas factor charges in cluster 5 point in opposite directions. We explain this linear trend of cluster grouping as a consequence of a trend of the chemical evolution from summit wells (cluster 1) with small recharge areas and shallow aquifers towards midslope (cluster 2) and footslope (cluster 5) passages with thicker overburden strata The latter are affected by minor surface influences and longer groundwater travel times/residence times within the bedrock. Data points of different aquifer storeys are grouped (but within the clusters), and slope position and aquifer depths seem to be more important for groundwater chemistry than the position in hydrostratigraphy, here. Apart from this trend, cluster 3 which is limited to the deepest well (in footslope position) exhibits high factor charges in $Ca^{2+}$, $SO_4^{2-}$, EC_25, $Sr^{2+}$ and low charges in $Ba^{2+}$, $HCO_3^-$, $Fe^{3/4+}$, $Mn^{2/4+}$, whereas cluster 4 (limited to shallow wells in midslope position) shows opposite factor charges for these parameters. In general, clusters are grouped in a line of chemical development and by oxic/anoxic conditions and different mineralizations depending on the position in aquifer stratigraphy.

**5.1.3 Modes of groundwater flow**

Within the aquifer-aquitard-sandwich, fast conduit groundwater flow, which is typical for karstified carbonate rocks (Wong et al., 2012), likely takes place in the moTK-1 and partially in the moM-1 aquifer storeys, whereas slow diffusion in slightly fractured, thin aquifers beds is anticipated in the moM-2 to moM-9 aquifer storeys. Confined diffuse flow is considered laminar and takes place in interparticle pores and fractures of dense limestones with low primary porosity (Smart and Hobbs, 1986). This results in the well oxygenated moTK-1 and moM-1 groundwaters and a significant oxygen consumption/deficiency (coupled to the mobility of $Fe^{2+}$ and $Mn^{2+}$-ions and the low mobility of $NO_3^-$ and $SO_4^-$ ions) in the moM-2 to 9 groundwater, resulting in completely different milieu conditions for the biogeochemical processes and for the life in the subsurface as well.

## 5.2 Surface-subsurface connectivity

Potential zones for interactions between surface and subsurface waters are the outcrop areas of the aquifers, incised valleys (cutting into the stratigraphic succession), springs and surface karst phenomena. Nearby faults and valleys, an increased number of additional non-stratabound, connected fractures that provide pathways for pronounced and preferential fluid flow (Meier et al., 2015) are likely.

### 5.2.1 Preferential recharge areas

As a typical feature of aquifer-aquitard alternations in a tilted hillslope setting, preferential recharge takes place at the aquifer outcrop zones (Andreu et al., 2011; Fig. 2 A). Based on the tectonically tilted and subsequently exhumed hillslope, the main aquifer recharge area is situated in the upper hillslope/summit area, which is covered by forest with very low anthropogenic impact. A predictable geological structure allows a general tracking of flow paths from the recharge to the discharge area. The second route for preferential recharge is related to the lines of caprock sinkhole lineaments, which can be tracked over more than four kilometers in the midslope between transect locations H2/3 and H4. The origin of these sinkholes does not lie in the karstification of the Upper Muschelkalk strata itself, as the combination of aquifer fracture networks (tight conduits) and stabilizing, insoluble aquitard beds do not cause sufficient mass deficits that will allow hanging wall collapses. Here, caprock sinkholes are related to mass deficits by subrosion in the underlying evaporite rocks with gypsum and halite (Mempel, 1939; Malcher, 2014). As caprock sinkholes are arranged in lineaments (Fig. 2 A) parallel to regionally known fault orientations, it is very likely that they are coupled to the penetration of surface/subsurface water at fracture zones (compare to: Smart and Hobbs, 1986; Worthington, 1999; Klimchouk, 2005), that promote preferential recharge (Smart and Hobbs, 1986; Suschka, 2007). Although dissolution of soluble rocks is limited within the target aquifers of this study, it is reasonable, that collapse structures are accompanied by enhanced rock fracturing and permeability.

### 5.2.2 Influences of soils in the recharge area

The spatial distribution of the soil groups reflect the outcrop zones of limestones/marlstones (Greitzke and Fiedler, 1996; Brandtner, 1997; Rau and Unger, 1997) and the succession of aquifer storeys in the summit/shoulder area of the hillslope which exhibits little quarternary rocks coverage and thin soils (Rendzic Leptosol, Chromic Cambisol) with favorable infiltration properties in the aquifer outcrop areas of moTK-1 and moM-1/6/8. Thicker relictic Chromic Cambisols which are formed by intensive decarbonatization (Rau and Unger, 1997; AG Boden, 2005) are restricted to accumulations in former depressions (i.e. caprock sinkholes) on shoulder regions of the hillslope (moTK-1 and moM-1 outcrops). Besides these depressions, Chromic Cambisols with low hydraulic conductivities are typically the subsoil layer of sequences with two layer superimposed soils, resulting in lateral soil interflow (Ali et al., 2011) as it is observed in the shoulder and midslope region during the measurement of soil hydraulic conductivities. Soils on unfractured marlstone/claystone aquitards are typically Calcaric Regosols and Stagnosols with low hydraulic conductivities of both, soils and parent rocks. According to our dataset, soil thickness is primarily controlled by the slope gradient of transverse valleys with increasing soil thickness and water storage options towards the center of valleys. As a result of the increasing coverage of parent rocks by loess loam from the regional midslope to the footslope, (related to solifluction of windblown dust; Kleber, 1991; Bullmann, 2010), loess loam becomes the dominating soil substratum in the outcrop areas of aquifer storeys moM 7/8/9. In this area, Luvisols cover local ridges and Luvisol-Stagnosol-transitions (related to the continuous vertical clay relocation (Rau and Unger, 1997) cover local valleys, whereas older caprock sinkholes are either filled by colluvisols or with loess loam, leading to either poor or very good infiltration/recharge properties. In the same context, Bachmair et al. (2009) considers the role of micro-depressions in soils as one of the major factors for preferential recharge.

Although median grain size, grain size sorting and the standard deviation of grain size fractions is comparable in all soils,
"carbonate series soils" (Rendzic Leptosol and Chromic Cambisol) offer lower hydraulic conductivities than "siliciclastic
series soils" (Cambisol, Luvisol, Stagnosol) by one order of magnitude. Soil hydraulic conductivities (Ks) are uncorrelated
to soil texture properties. For this reason a linear transfer function (Ks vs. median grain size or grain size category) for
extending the 16 Ks measurements to the 271 mapped soil profiles was not possible with the actual dataset. A low
correlation between Ks and texture is related to strong influences of soil structure and aggregation (Totsche et al., 2017)
rather than texture on Ks. Structural parameters are for instance the content and proportion of macro and microaggregates, a
hierarchical aggregate system, the presence and frequency of secondary pores and anthropogenic changes like former plough
traces or traces of forestry machines. Bachmair et al. (2009) for instance identify tillage-related macropores as a main factor
for deep infiltration into cropland soils, whereas surface littler layers inhibit infiltration in forest soils. Moreover preferential
infiltration through biomacropores (i.e. earthworm borrows, root channels), that bypass the soil matrix are of great
significance in aggregated and argillaceous soils with low matrix conductivity for the infiltration during heavy rain events
(Deurer et al., 2003; Weiler and Naef, 2003; Blouin et al., 2013). Klaus et al. (2013) identify vertical macropores of anecic
earthworms as a major flow control in a hillslope tile drain system. Wienhöfer and Zehe (2014) additionally consider loose,
litter-rich topsoils, lateral preferential pathways, preferential pathways at the soil-bedrock surface and bedrock topography.
**5.2.3 Influences of land management in the recharge area on groundwater quality**
The forest areas as a source of groundwater from catchment-near summit/shoulder wells are confirmed by their aquifer
outcrop areas (moTK-1, moM-1) within the managed (and partly unmanaged) forest. In contradiction, we detected
potentially agriculture-related substances ($NO_3^-$, $K^+$, $Cl^-$) in the wells drilled to all aquifer storeys in midslope location wells
(H31/41), although, these groundwater types are recharged mainly within the forest (up to 96.5 % forest). These surface
signals are interpreted to be related to the cropland and village areas within the preferential recharge zones with sinkhole
lineaments. Anoxic groundwater in aquifers (moM8/9) partially recharged from outcrop areas with agricultural and village
land use, are likely attributed to microbial oxygen depletion resulting from the degradation of organic carbon or oxidation of
inorganic electron donors. For shallow wells in valleys (H42/43/4S), lateral soil water inflow with high organic/fertilizer
load towards the valley provide an alternative way for the enhancement of oxygen depletion. Much lower $NO_3^-$, $K^+$, $Cl^-$
concentrations in the footslope wells (H52/53) are likely a consequence of the argillaceous caprocks that increase in
thickness towards the footslope.
**5.2.4 Utilization of the recharge potential for interpreting groundwater quality**
Due to the recharge potential map (Fig. 8), thick, limestone-dominated aquifer storeys in the deeper sections of the
hydrostratigraphy (i.e. moTK-1) are mainly recharged in their outcrop areas and subordinarily in sinkhole lineaments and
transverse valleys. Within these areas, Ah-Cv soils (i.e. Rendzic Leptosol) and fractured limestone bedrocks offer the highest
recharge potential. A drastic decrease in recharge potential with increasing overburden strata is confirmed by low
concentrations in surface-related substances (i.e. $NO_3^-.Cl^-$, $K^+$) in well sites H1/2/4/5 for the aquifer storey moTK-1.
Increasing concentrations of ions, which are related to the dissolution of the carbonate (aquifer) bedrock ($Ca^{2+}$, $Mg^{2+}$, $HCO_3^-$
and $Sr^{2+}$) point to a chemical evolution of groundwater in the direction of discharge (H1 to H5). A moderate to high recharge
potential combined with cropland/pasture areas) in the recharge direction of site H3 is reflected by increased concentrations
in TOC, $NO_3^-.Cl^-$, $K^+$, $Na^+$ and $O_2$ (Fig. 6 and 8). Aquifer storeys with thin-bedded limestone-marlstone alternations (i.e.
moM-8, Fig. 8) generally show a lower recharge potential in their aquifer outcrop zones, compared to thick, limestone-
dominated aquifers (moTK-1), which is related to thicker soils and a high marlstone-limestone ratio. This is reflected in the
groundwater quality of moM-8 wells (H42/43/53) with low Eh and low concentrations in $O_2$, pointing to slow flow velocities
and long residence time within the argillaceous soils and marlstone-dominated and thinly fractured bedrocks.
**5.2.5 Flow directions**
The four modes of presumed groundwater flow are (1) vertical percolation, (2) bedding-parallel flow, (3) descending, cross-
formational flow and (4) ascending cross-formational flow. (1) Vertical percolation through the soils and the unsaturated
zone takes place in the whole Hainich CZE catchment area, following diffuse/non-point infiltration. As the aquitard
interbeds highly reduce the vertical flow connections, outcrop zones of the aquifer storeys as well as caprock sinkholes act as
important preferential infiltration pathways which are typical for hillslope recharge zones. Since caprock sinkholes remain
dry, even directly after precipitation events, high infiltration rates can be assumed for these structures. (2) An undisturbed
sequence of aquifers and aquitards with constant bed characteristics, comparable groundwater chemistry within the same
aquifer storey and the high pressure heads (well H51/52/53: 70/50/20 m rise within the well pipe) are interpreted to indicate
bedding-parallel flow (stratigraphic flow control; Goldscheider, 2005) in the groundwater transit zone. This is supported by
the spatial variation of groundwater chemistry within the same aquifer storey (i.e. increase in $K^+$, $Na^+$, $Mg^{2+}$ and $Si^{4+}$ within
moM-8 from well site H4 to H5), provided a cut-off from surface influences, there. Also the combination of saturated
conditions within the shallow storey and temporarily unsaturated conditions within the lower aquifer storey in well site H3
points to highly reduced vertical flow. Bedding parallel flow is also supposed for fractures with small fault displacements
that do not exceed the thickness of the interbedded aquitard (compare to Goldscheider & Drew, 2007). (3) Zones with
enhanced fracture indices (fractures/m drill core) in well sites H3 as well as consistent Fe/Mn-oxide fracture walls in all
aquifer storeys of this site points to a quick and cross-formational descending flow of oxygenated groundwater (thus with
low Fe/Mn mobility; Hem, 1985) via vertical master joints (term: Dreybrodt, 1988; Ford and Williams, 2007) close to
potential fracture zones. This is supported by higher concentrations in $Na^+$, $K^+$, $NO_3^-$, $Cl^-$ and TOC in the deeper site of H3
compared to the shallower well of this site, related to agriculture and fertilizing (Matthess, 1994; Kunkel et al., 2004). Cross-
formational descending flow in shattered or fractured rocks is assumed to take place in fracture zones (Worthington, 1999;
Goldscheider & Drew, 2007), tracked by lineaments of caprock sinkholes (Mempel, 1939; Hoppe, 1962; Smart and Hobbs,
1986; Jordan and Weder, 1995). (4) Groundwater chemistry of footslope well H51 (screen base at the base of moTK-1
aquifer) bears a combination of highly concentrated in $Ca^{2+}$, $SO_4^{2-}$, $Sr^{2+}$, $Cl^-$ (and $SO_4^{2-}/HCO_3^-$-ratios) and low concentrations
in $K^+$, $Na^+$, $NO_3^-$ concentrations that points to a non-surface import of sulphate (-chloride) groundwater due to dissolution of
evaporite rocks (Edmunds and Smedley, 2000). The location of this lowermost aquifer storey directly above dolomite,
gypsum and halite beds (Middle Muschelkalk; Jordan and Weder, 1995) points to ascending cross formational groundwater
flow (Garleb, 2002; Völker and Völker, 2002). The ascent of groundwater which is well known for artesian settings
(Klimchouk, 2005), could be related to hydraulically confined conditions as measured for site H5 and the assessed
groundwater flow crossing a sinkhole lineament between well site H4/H5 (Fig. 10). Alternatively, the ascent of Middle
Muschelkalk-groundwaters is probably related to a decrease in permeability within the Middle Muschelkalk aquifers, as
subrosion of sulfate rocks (succeeding from the summit to the footslope region) has not reached the groundwater transit zone
yet, resulting in the presence of low permeable sulfate rocks and an ascent of groundwaters discharging the Middle
Muschelkalk (Hoppe and Seidel, 1974; Garleb, 2002).
**5.3 Controls on groundwater quality and groundwater provenance**
Cluster 1 comprises the shallow to intermediate deep moTK-1 (H13/21) and moM-1 (H14) wells with a base depths of
screen sections between 7.0 and 30.5 m. Groundwater chemistry with high $HCO_3^-$ and $Ca^{2+}$ as well as low $Mg^{2+}$, $SO_4^{2-}$, $Na^+$,
$K^+$, $Si^{4+}$ concentrations (Fig. 8) correspond with the pure calcite limestone ($CaCO_3$) aquifer lithology. Low concentrations in
$Cl^-$, $K^+$, $Na^+$, $Mg^{2+}$ are related to parts of the catchment area that is exclusively used as forest and to soils (mainly Rendzic
Leptosols) with high hydraulic conductivities and with little clay content, resulting in short residence times of water in soils.
Relatively high concentrations in TOC are interpreted as little organic degradation due to short travel times. Fast infiltration
also arises from aquifer outcrop zones in culmination areas, allowing a high ratio of infiltration vs. lateral soil interflow.
According to the PCA, Cluster 1-wells could be interpreted as one end member at the beginning of a "groundwater
development series". Generally, groundwater compositions in summit-near regions of hillslopes are comparable to the
composition of the bedrock. Surface signals (i.e. soil or vegetation-derived organic carbon) can be traced more clearly due to
little organic degradation (and low filtering) along the flow path by the soils and bedrock pores/fractures.
Cluster 2 encompasses moderately deep moTK-1 wells (H31/41) and one moM-5/6 well (H32) with base depths of screen
sections between 22.0 and 47.5 m. The hydrochemistry with moderate concentrations in $Ca^{2+}$, $Mg^{2+}$, $SO_4^{2-}$ and low $K^+$, $Na^+$
as well as high Mg/Ca-ratios is not linked to the pure calcite limestones (moTK-1) or the calcite, illite/kaolinite/chlorite
mineral composition (moM-5/6) and requires an external source of $Mg^{2+}$. Nitrate and TOC contents are higher than expected
by the forest (moTK-1)/forest-dominated type of land management (moM-5/6) in the aquifer outcrop zones and are likely
explained by cropland/pasture in a sinkhole lineament in the recharge zones of sites H3/4. Oxygenated groundwater in both
aquifer storeys (thus with low Fe/Mn mobility; Hem, 1985) and a high degree in rock fracturing and fracture mineralization
with Fe-/Mn-oxide minerals (aquifer storeys moM-9/8/7 and 6), point to a vertical penetration with near surface groundwater
via fracture zones through all aquifer storeys. Enhanced concentrations in $Na^+$, $K^+$, $NO_3^-$, $Cl^-$ and TOC (Fig. 7) are likely
related to agriculture and fertilizing (Matthess, 1994; Kunkel et al., 2004) around the sinkhole lineaments. As nitrate, which
is generally derived from agricultural fertilizers (Agrawal et al., 1999; Jeong, 2001), is still present in the deep aquifer waters
of site H3, vertical bypassing through master joints (term: Dreybrodt, 1988; Ford and Williams, 2007), must be faster than
the denitrification process. Quick infiltration is here mostly related to the preferential sinkhole recharge, as the soils
(Luvisol/Cambisol, both with low-conductive subsoils and a high degree in lateral soil interflow) in the outcrop zones bear
only moderate to poor soil hydraulic conductivities. In general groundwater in the discharge of exceptionally highly
conductive preferential recharge spots can reflect surface signals, even in stratified aquifer/aquitard successions. These zones
are therefore suggested to be of uppermost importance for groundwater protection.
Cluster 3 contains groundwater of the deepest moTK-1 well (screen base 88 m) drilled to a cropland plot in the footslope
area. As it is discussed in 4.2.5, ascending sulphatic groundwater is likely responsible for the $SO_4^{2-}$ - $HCO_3^-$ type of
groundwater. High chloride concentration is interpreted with a long groundwater travel distance. Very weak surface signals
(in form of low nitrate and TOC concentrations) attest the inhibition of vertical infiltration by aquitards and a likely result
from long groundwater travel distances (high residence time) from the forest recharge zone of the moTK-1 aquifer.
However, high groundwater oxygen concentration in more than 5 kilometers from the capture zones requires a very quick
oxygen supply by more rapid flow events. In general, deeper groundwater in the transit zone may bear surface signals, if
flow velocities are great enough for transporting young groundwater into the subsurface. This is applied for the more
karstified aquifer storeys in aquifer/aquitard successions.
Cluster 4 includes groundwater of the shallow moM-6/7/8 wells H42/43/H4SB with screen depths in 11.5 to 19.0 m. Low-
conductive alluvial cover sediments reduce vertical infiltration drastically. High $HCO_3^-$ and low $Na^+$, $K^+$, $Si^{4+}$, $Sr^{2+}$
concentrations as well as the moderately high Mg/Ca ratio in the groundwater reflect the calcite and dolomite aquifer
mineralogy. High $Fe^{(2+\text{ and } 3+)}$, $Mn^{(2+\text{ and } 4+)}$ and low $SO_4^{2-}$ concentrations correspond to a low redox potential and the absence
of dissolved oxygen, controlling the mobility of these ions (Hem, 1985; Hsu et al., 2010). Although the aquifer catchment of
moM-6/7/8 is of mixed forest/agriculture/village type, TOC and $NO_3^-$ concentrations are low, due to the consumption of
organic matter and the denitrification under anoxic conditions (Agrawal et al., 1999). Relatively high $Cl^-$, $K^+$ concentrations
are probably remnants of agriculture/village-related surface signals. The discrepancy of a close distance to the recharge area
and the low oxygen content could be explained by low hydraulic subsoil conductivities of moM-7 outcrop area (Luvisols,
Stagnosols, Cambisols) and very narrow fractures in the aquifer rocks of moM-8. Generally aquifer confinements in valleys
of hillslope aquifers could result in low aquifer quality, directly depending on the fracture sizes and soil hydraulic
conductivities.
Cluster 5 encompasses groundwater from deep (50.0 - 69.0 m screen depths) aquifers storeys moM-3/4/8 (wells H52/53)
which are covered by very thick (> 40 m), low permeable cover strata (Erfurt formation, Warburg formation). High $K^+$,
$Na^+$, $Mg^{2+}$ and $Si^{4+}$ concentrations correspond to long residence times of groundwater (Bakalowicz, 1994; Khan and Umar,
2010) in marlstone/limestone successions due to the distant (>3 km) recharge zones. If the grouping of groundwater types in
the PCA represents a "chemical evolution series", then cluster 5 stands for the most developed /oldest groundwater. In
general, caprock-covered footslope aquifers with narrow fractures contain likely more developed groundwater that mostly
reflect subsurface influences and less surface influences (i.e. in comparison with more conductive aquifers of the same well
site H5).
**5.4 Assessment of vulnerability factors**
The configuration of the aquifer system in the hillslope setting also controls the groundwater resource vulnerability. With our
multi-method investigation of the different subsurface compartments, we also revealed factors of the areas´ intrinsic
vulnerability. Characteristics of intrinsic vulnerability are solely controlled by hydrogeological properties of the aquifer and
overburden (Vrba and Zoporozec, 1994), and integrate the inaccessibility (i.e. by low-permeable cover strata) of the
saturated zone and the attenuation capacity (retention, turnover) of the overburden (Adams and Foster, 1992).
The stacking of aquifers/aquitards, the coverage with caprocks and the lateral continuity of strata reduces the overall
dominating disperse infiltration and, thus the intrinsic vulnerability. Threatening of single aquifer storeys or assemblages is
predominantly controlled by fractures/faults (valleys) or karst phenomena bypassing the protective cover of soils and
unsaturated zones (Fig. 8). A moderate degree of physical filtering by the narrow fractures, which are the predominant flow
paths, is assumed. Also the claystone and marlstone interlayers bear a certain filtering of contaminants by retention. The
preferential recharge/outcrop zones of the aquifer storeys are characterized by generally highest vulnerability. However,
these zones, located in the summit to upper midslope of low mountain ranges, are mostly covered by forest. Generally, the
summit position of outcrop zones of the main aquifer storey (moTK-1) lowers the risk for contamination, paradoxically, due
to the thin soils that prevented lasting agricultural use and settlements. Further zones of higher vulnerability are located in
the discharge of sinkhole lineaments or fracture zones that bypass surface water or drains directly into the aquifers. A
mapping and structural investigation of these geostructural links will be recommended for (i.e.) proper dimensioning of
drinking water protection zones. In comparison to classical karst sites (i.e. massive carbonates) with pronounced karst
phenomena, (Doerfliger et al.,1999, Witowski et al., 2002), our portrayed setting shows an overall low to moderate
vulnerability, that is locally elevated by inherent factors (outcrop zones) and inherited features (i.e. karstification of the
underlying Middle Muschelkalk subgroup).
**6 Conclusions**
We applied a multi-method approach for a detailed investigation of Critical Zone functioning and reconstruction of
groundwater quality in the hitherto scarcely described setting of thin-bedded alternating carbonate-/siliciclastic rocks in a
hillslope terrain. For the Hainich Critical Zone Exploratory that offers unique access to a multi-storey groundwater system in

the common and widely distributed geological setting, we found the following factor manifestations of local geology, relief, soils and land use for recharge and chemical evolution of groundwater:

- Low-permeable marlstone beds within a marine succession of high lateral continuity represent a number of aquitards that cause a multi-storey hydrostratigraphy of the Upper Muschelkalk formations.

- As a multi-storey hydrostratigraphy exhibits limited vertical percolation, the outcrop zones of dipping aquifer storeys become very important as preferential surface-recharge areas for inputs of matter and energy.

- Diffuse fracture flow dominates over karst/conduit flow in the mixed/multi-layered lithology. Subsurface water flow predominantly takes place in bedding-plane parallel mode / in stratabound fractures of the limestone beds and it is trackable from the recharge areas along the storeys.

- From summit to footslope positions, travel distances and presumably groundwater ages generally increase. For the individual storeys however, travel distances to monitoring wells decrease in the downslope direction, whereas their groundwater ages very likely increase due to lower fracturing and higher retention. In the same direction, surface controls (i.e. nutrient input) decrease and subsurface controls (water-rock-interaction) increase.

- Compared to more vulnerable settings (i.e. massive carbonate karst, open karst), the mixed carbonate-siliciclastic alternations exhibit moderate intrinsic vulnerability. This is due to lateral continuity of low permeable interbeds, soil covers and caprocks, of which the latter successively increase in thickness towards the footslope. Areas downstream the caprock sinkhole lineaments (and likely transverse valleys) are likely more threatened by anthropogenic (mostly agricultural) inputs.

- The quality of groundwater resources with peripheral hillslope recharge benefits from extensive land management or, ideally (managed/unmanaged) forest coverage and reveals the importance of recharge area protection.

In general, for mixed carbonate-/siliciclastic rock alternations that are prone to develop multilayered aquifer systems, both the aquifer configuration (spatial arrangement of strata, hillside cutting, outcrop positions) and the related geostructural links (preferential recharge areas, karst phenomena) are major controls of impacting surface and subsurface factors. For the studied type of thin-bedded carbonate aquifer setting, we could to demonstrate, that a comprehensive investigation of aquifer connectivity in the transit/discharge area as well as soil cover and land use in the recharge area is mandatory and must be rated indispensable for a thorough understanding of the state and evolution of groundwater quality. Linking groundwater hydrochemistry mostly to surface factors such as land use would result in a contradiction, as groundwater chemistry does not reflect the type of land use in immediate proximity to the wells. Furthermore, footslope wells´ hydrochemistry is strongly impacted by aquifer stratigraphy, karst phenomena input and the cross-formational ascent of sulphatic groundwater that could not be evaluated without spatial lithostratigraphical data. A geostructural investigation and mapping is essential for the localization of aquifer outcrop areas and the assumption of stratigraphy-controlled flow directions. In case the soils group and the type of land use would have been neglected, discrimination between influences by natural and anthropogenic controls would have been ambiguous. If the dataset included hydrochemistry and multivariate statistics only, the interpretation of this dataset would have been ambiguous, also as different chemical surface/subsurface sources result in similar hydrochemical compositions.

Recent studies of the CRC AquaDiva focus on signal transit and transformations of surface signals. This is for instance applied to surface-sourced organic matter and microorganisms (Küsel et al., 2016, Schwab et al., 2017, Lazar et al., 2017) to further investigate subsurface connectivity, surface-subsurface interactions and functions of microbial life in groundwater environments. Further CZ exploration should also aim the investigation of deeper strata connection by regional groundwater flow, hydraulic properties and proportions of unsaturated zones and matter processing within.

## Data availability

Plot data will be deposited on the BExIS2 (http://bexis2.uni-jena.de) data portal of the CRC AquaDiva (https://aquadiva-pub1.inf-bb.uni-jena.de). Access to the AquaDiva data portal for review purposes is granted. Hydrochemical data will not be shared publically at this point of time, as this paper represents the first of a series of three related papers, using similar data sets for different aims. For instance the prepared time series analysis as well as flow and transport models will rely on the same data sets.

## Team list

Bernd Kohlhepp (BK), Robert Lehmann (RL), Paul Seeber (PS), Kirsten Küsel (KK), Susan E. Trumbore (SET) and Kai U. Totsche (KUT).

## Author contribution

BK and RL contributed equally to this article. RL and BK organized and BK conducted the geological mapping, soil mapping, as well as the analysis and correlation of drill cores and geophysical logs. RL organized and partly conducted the groundwater monitoring and obtained authorization from authorities and landowners/companies. PS carried out parts of the soil mapping/description and hydrochemical analyses and carried out the descriptive and multivariate statistics. KUT, SET and KK provided editorial comments on the manuscript. KUT designed and coordinated the project and supervised the field work, data analysis, data presentation and manuscript preparation. KUT, KK and SET are speakers of the CRC AquaDiva.

## Competing interests

The authors declare that the research was conducted in the absence of any commercial or financial relationships that could be construed as a potential conflict of interest.

## Acknowledgements

The work has been (partly) funded by the Deutsche Forschungsgemeinschaft (DFG) CRC 1076 "AquaDiva" and the state of Thuringia 'ProExzellenz' initiative AquaDiv@Jena (107-1). Field work permits were issued by the responsible state environmental offices of Thuringia, local authorities and landowners. We thank Christine Hess and Maria Fabisch for scientific coordination. Heiko Minkmar supported the field surveys and conducted water and ramcore samplings. Tanja Reiff conducted the Guelph permeameter measurements. Sören Drabesch and Marie Mollenkopf constructed the conceptual soil map. Matthias Händel carried out the FT-IR and XRD measurements on bulk rock samples.

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

  **Tables**

  **Table 1** Soil groups (WRB, 2003 and AG Boden, 2005), soil texture and soil hydraulic properties

| WRB soil group | Rendzic Leptosols, Calcaric Regosols | Chromic Cambisols | Cambisols | Luvisols | Stagnosols | Fluvic Cambisol |
|---|---|---|---|---|---|---|
| German soil group | Rendzina, Pararendzina | Terra fusca | Braunerde | Parabraunerde | Pseudogley | Vega |
| Substratum | Limestone, marlstone | limestone | marlstone, loess loam | loess loam, marlstone | loess loam, marlstone | alluvial silt+clay |
| Thickness | 5-35 cm | 85-100 cm | 35-80 cm | 50-100 cm | 80-190 cm | up to 3.3 m |
| Soil category | medium silty clay | slightly silty clay | medium silty clay | slightly clayey silt | medium silty clay | strongly silty clay |
| Median grain size (µm) | $12.6 \pm 3.6$ | $13.3 \pm 4.9$ | $13.2 \pm 3.6$ | $17.3 \pm 4.2$ | | $17.8 \pm 5.0$ |
| % Clay + Fine silt | $34.0 \pm 7.8$ | $33.9 \pm 9.7$ | $32.3 \pm 7.0$ | $27.0 \pm 9.4$ | | $25.6 \pm 9.1$ |
| Water storage capacity | low | high | high | medium | very high | very high |
| Roots | < 35 cm | 30-60 cm | 20-80 cm | 20-180 cm | 30-80 cm | 15-40 cm |
| Cracks and borrows | earthworm borrows down to the host rock | earthworm borrows down to the host rock | frequent | deep borrows (voles, worms) | limited to shallow soil horizons | limited to shallow soil horizons |
| Decarbonatization | no | complete | almost complete | complete | incomplete | no |
| Typical color (subsoil) | yellowish gray | dark yellowish gray | brownish gray | yellowish gray | greenish gray | yellowish gray |
| | 10YR3/2 | 7,5YR4/3 | 10YR5/6 | 10YR4/6 | 2,5Y4/2 | 10YR4/3 |
| Hydromorphic attributes | no | oxidative | oxidative | oxidative | oxidative + reductive motteling | oxidative + reductive motteling |
| Soil water | > 200 cm | rarely and > 80 cm | rarely and > 100 cm | rarely and > 100 cm | commonly in 1-2 m | very common > 40-60 cm |
| Morphologic position | middle slope + all steep local slopes | culmination + local plateaus | culmination to middle slope | lower slope, (middle slope) | lower slope, valley | valley center |
| Land use | forest + military training area | forest + some grassland | cropland, grassland, forest | cropland, grassland, uncommonly forest | grassland, uncommon-ly forrest | grassland |
| Anthropogenic changes | compacted (tanks), drained | uncommon | ploughed, compacted | ploughed, compacted | drained | drained |
| Hydraulic con-ductivity Ks [m/s] (Median, ± SD) | $5.2*10^{-5}$ $\pm$ $4.9*10^{-5}$ | $2.5*10^{-5}$ $\pm$ $1.2*10^{-5}$ | $1.5*10^{-4}$ $\pm$ $5.2*10^{-5}$ | $1.2*10^{-4}$ $\pm$ $1.2*10^{-4}$ | $6.5*10^{-7}$ | not measured |

**Table 2** Hydrostratigraphic standard section: relative positions of aquifer storeys (this study) in comparison with a published classification (Küsel et al., 2016) in the context of the the German stratigraphy and time scale.

| Time scale (DSK, 2002; Gradstein et al., 2003) [Ma] | Level above base of moTK [m] | German stratigraphy (DSK, 2002) | Regional stratigraphy (Central Germany. Seidel, 2003) | Aquifer assemblages (Küsel et al., 2016) | Aquifer storeys (this study) X aquifer - aquitard | | Level above base of moTK [m] | Ground-water well (screen section) |
|---|---|---|---|---|---|---|---|---|
| 0.01...0 | | | soils alluvial sediments (qhf) | | | | | |
| 0.1…0.01 | | | loess (qwLo) | | | | | |
| 232.5...0.1 | | | erosional unconformity | | | | | |
| 235.0...232.5 | 61.3… >90.0 | Erfurt formation (kuE) | Graue Mergel (kuGM) | | - | | | |
| | | | Sandstein-komplex1 (kuS1) | | X | **kuS1-2** | 74.5…76.5 | |
| | | | | | X | **kuS1-1** | 68.0…70.0 | |
| | | | Grenzschichten (kuGR) | | - | | | |
| | 43.4… 61.3 | Warburg formation (moW) | Glasplatten (moCGP) | | - | | | |
| | | | Glaukonitbank (moCG) | | X | **moW-1** | 53.0…54.0 | |
| | | | Zinkblendebank | | - | | | |
| | | | Fischschuppen-schichten (moCFU) | | - | | | |
| | | | | | X | | | |
| | | | Cycloidesbank (moCC) | | X | **moM-9** | 42.0…45.0 | |
| 238.5...235.0 | 8.7… 43.4 | Meissner formation (moM) | Discites-schichten (moCD) | **HTU** (Hainich transect upper aquifer assem-blage) | X | **moM-8** | 38.0…40.5 | H42, H43, H53 |
| | | | | | - | | | |
| | | | | | X | **moM-7** | 34.0…36.0 | H4SB |
| | | | | | - | | | |
| | | | | | X | **moM-6** | 31.0…33.7 | H32, H4SB |
| | | | | | - | | | |
| | | | | | X | **moM-5** | 27.0…29.0 | H32 |
| | | | | | - | | | |
| | | | | | X | **moM-4** | 22.9…24.5 | H52 |
| | | | | | - | | | |
| | | | | | X | **moM-3** | 18.9…21.8 | H52 |
| | | | | | - | | | |
| | | | Gervilleien-schichten (moCGV) | | X | **moM-2** | 15.2…18.3 | H23 |
| | | | | | - | | | |
| | | | | | X | **moM-1** | 8.0…13.2 | H14 |
| | | | | | - | | | |
| >238.5 | 0.0… 8.7 | Trochiten-kalk formation (moTK) | Trochitenkalk (moT) | **HTL** (lower aquifer assem-blage) | X | | | H11, H12, H13, H21, H22, H31, H41, H51 |
| | | | | | | **moTK-1** | 0.0…6.6 | |

**Figures**
**Fig. 1.** Location of the Hainich CZE (a + b): Prominent karst springs in the Hainich CZE are coupled to NE-SW orientated
fault zones (b; modified from Mempel, 1939 and Jordan and Weder, 1995). (c): Geological setting of the eastern Hainich
hillslope with monitoring wells of the research transect, accessing Upper Muschelkalk target formations (mo); Data sources:
DEM ©GeoBasisDE/TLVermGeo, Gen.-Nr.: 7/2016.
**Fig. 2.** (A): Types of land management, outcrop zones of aquifer storeys, sinkhole lineaments, potential fracture zones and
measurement points for soil hydraulic conductivities. Aquifer storeys which are lower in stratigraphy outcrop in higher
positions on the Hainich hillslope. The AquaDiva well transect H1 to H5 (also shown) covers hillslope regions from the
summit (H1) to the footslope (H5). (B) Conceptional soil map showing calculated soil groups and mapped calibration data
points. Signatures of the mapped soil profiles represent the grain size class (soil category) of the topsoil. Interpolated isolines
of mapped soil thickness show increasing thickness towards the NE and towards the transverse valleys.
**Fig. 3.** Physical properties of topsoil/subsoils and parent rocks of the four major soil groups. (A): Median frequencies of the
fine grain size fractions (clay + fine silt) showing increasing proportions towards the bedrock, respectively. (B): Median soil
hydraulic conductivities of soil groups showing higher median soil hydraulic conductivity in Chromic Cambisols and
Luvisols compared to the Rendzic Leptosols and Cambisols and general decreases in hydraulic conductivity towards the
subsoil. The error bars describe the root of the estimation variance of the average.
**Fig. 4.** Graphical correlation of marlstone/claystone intervals in gamma-ray logs, biostratigraphic limestone marker beds
(grainstones/rudstones) and the degree of karstification (red bars and red scale bar below: solution-enlarged bedding planes
to karst breccia). The geological aquifer correlation is cross-checked with the hierarchical clustering of hydrochemical
parameters.
**Fig. 5.** Surface and subsurface properties of the aquifer outcrop zones (moTK-1 to moM-9) on the Hainich hillslope. Area
sizes are related to the 29 km$^2$ area of this study. (A): Absolute abundances of land management types in the preferential
recharge areas of the aquifer storeys. The two basal aquifer storeys are characterized by the largest aquifer outcrop areas and
the highest amounts of forest within these areas. Agricultural land management increases towards the higher aquifer storeys
(moM-2 to moM-9). (B): Relative abundances of soil groups within the aquifer outcrop zones showing high diversitiy in all
aquifer outcrop zones depending on slope positions and quarternary loess loam/alluvial clay coverage, (C): Grain size groups
of the topsoils and subsoils in the aquifer outcrop zones.
**Fig. 6.** Stratigraphic succession of the Upper Muschelkalk with stratigraphic marker beds (left), a gamma-ray log, aquifer
assemblages HTL/HTU, aquifer storeys moTK-1 to moM-9 and the average chemical compositions of monitored
groundwater (left: ions of carbonate/sulphate minerals; middle: redox-sensitive ions; right: ions which are potentially related
to the type of land management). The color code of hierarchical clusters is identical to Fig. 4.
**Fig. 7.** Principal component analysis (PCA) biplot for the complete parameter set (a) and for the limited parameter set
(b) without redox related parameters. Five clusters can be distinguished in both parameter sets. Samples within the clusters
are identical for both PCA plots. Factor loads for PC1 and PC2 are displayed as labels of the x/y-axis as well as the sequence
of factor loads for individual components. The color code of hierarchical clusters is identical to Fig. 4.
**Fig. 8.** Recharge potential maps for the two contrasting aquifer storeys moTK-1 (upper map) and moM-8 (lower map). The
recharge potential is a qualitative indicator for infiltration and percolation towards selected aquifer storeys. The brighter the
output color, the higher the recharge potential. Aquifer outcrop zones show highest recharge potentials, followed by sinkhole
lineaments and (NE-SW orientated) transverse valleys. Recharge potential decreases drastically with increasing thickness of
overburden strata towards the NE. Datapoints of field measurements are displayed in the left map (moTK-1 recharge
potential) and color-coded with respect to their soil hydraulic conductivity; Data sources: DEM ©GeoBasisDE/TLVermGeo,
Gen.-Nr.: 7/2016.
**Fig. 9.** Average groundwater chemistry of the hierarchical clusters 1-5 (left: groundwater wells and aquifer storeys within
the clusters). First column: parameters related to the carbonate-$CO_2$-equilibrum, $2^{nd}$ column: dissolved $O_2$, redox potential
and redox-sensitive ions, $3^{rd}$ column: ions which are potentially related to land management, $4^{th}$ column: ions related to the
dissolution of either carbonate/sulphate or clay minerals. The error bars describe the root of the estimation variance of the
average. The colour code of hierarchical clusters is identical to Fig. 4.
**Fig. 10.** Conceptual cross section of the Hainich hillslope showing the overall geological structure with the ten aquifer
storeys illustrating preferential aquifer recharge zones in the summit to midslope region as well as the discharge direction.
Lineaments of caprock sinkholes crossing the potential groundwater flowpath represent potential zones for descending or
ascending cross-formational flow and also the mixing of different types of groundwater.
**Fig. 1**

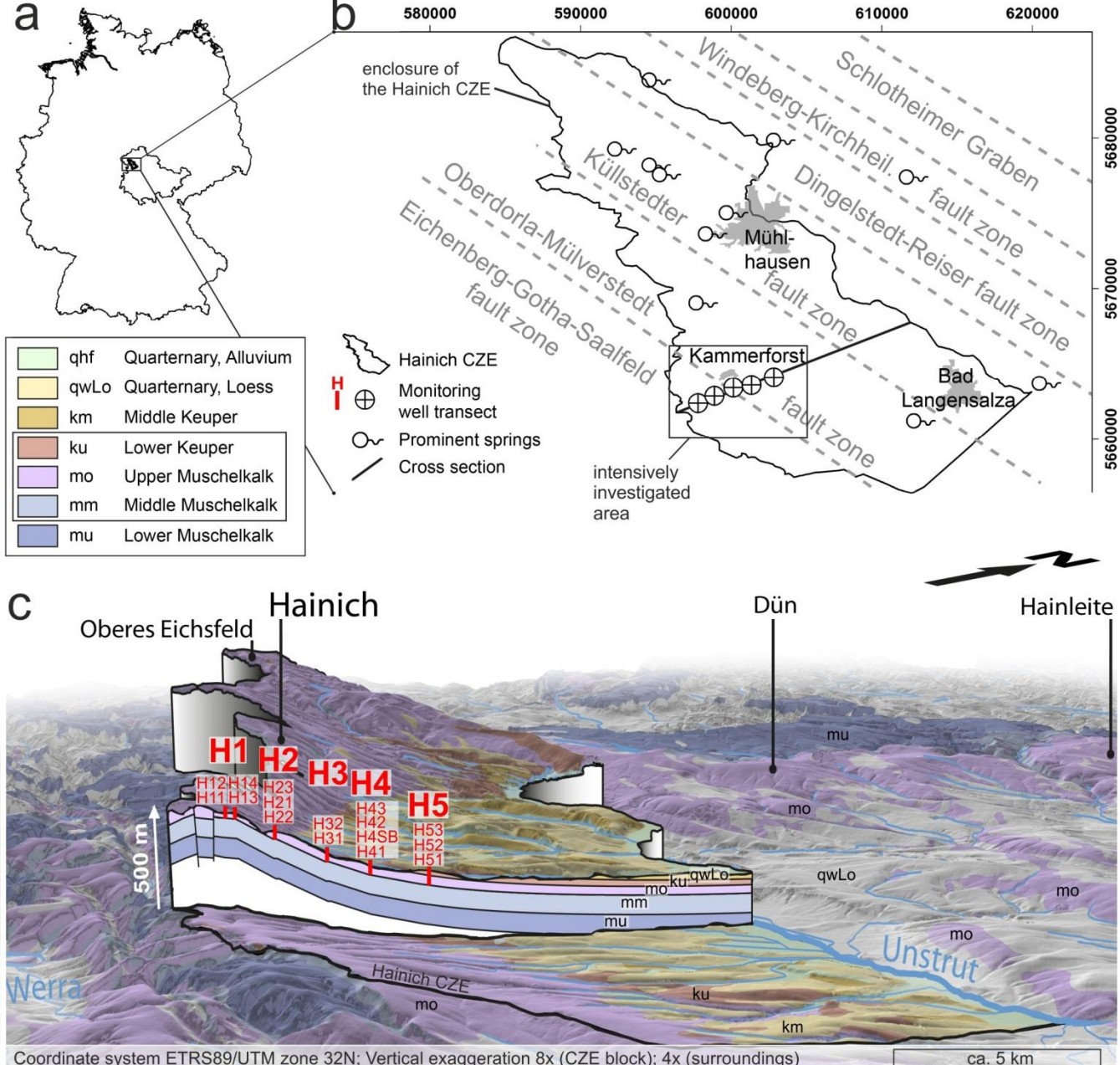

Coordinate system ETRS89/UTM zone 32N; Vertical exaggeration 8x (CZE block); 4x (surroundings)
**Fig. 1.** Location of the Hainich CZE (a + b): Prominent karst springs in the Hainich CZE are coupled to NE-SW orientated
fault zones (b; modified from Mempel, 1939 and Jordan and Weder, 1995). (c): Geological setting of the eastern Hainich
hillslope with monitoring wells of the research transect, accessing Upper Muschelkalk target formations (mo); Data sources:
DEM ©GeoBasisDE/TLVermGeo, Gen.-Nr.: 7/2016.
**Fig. 2**

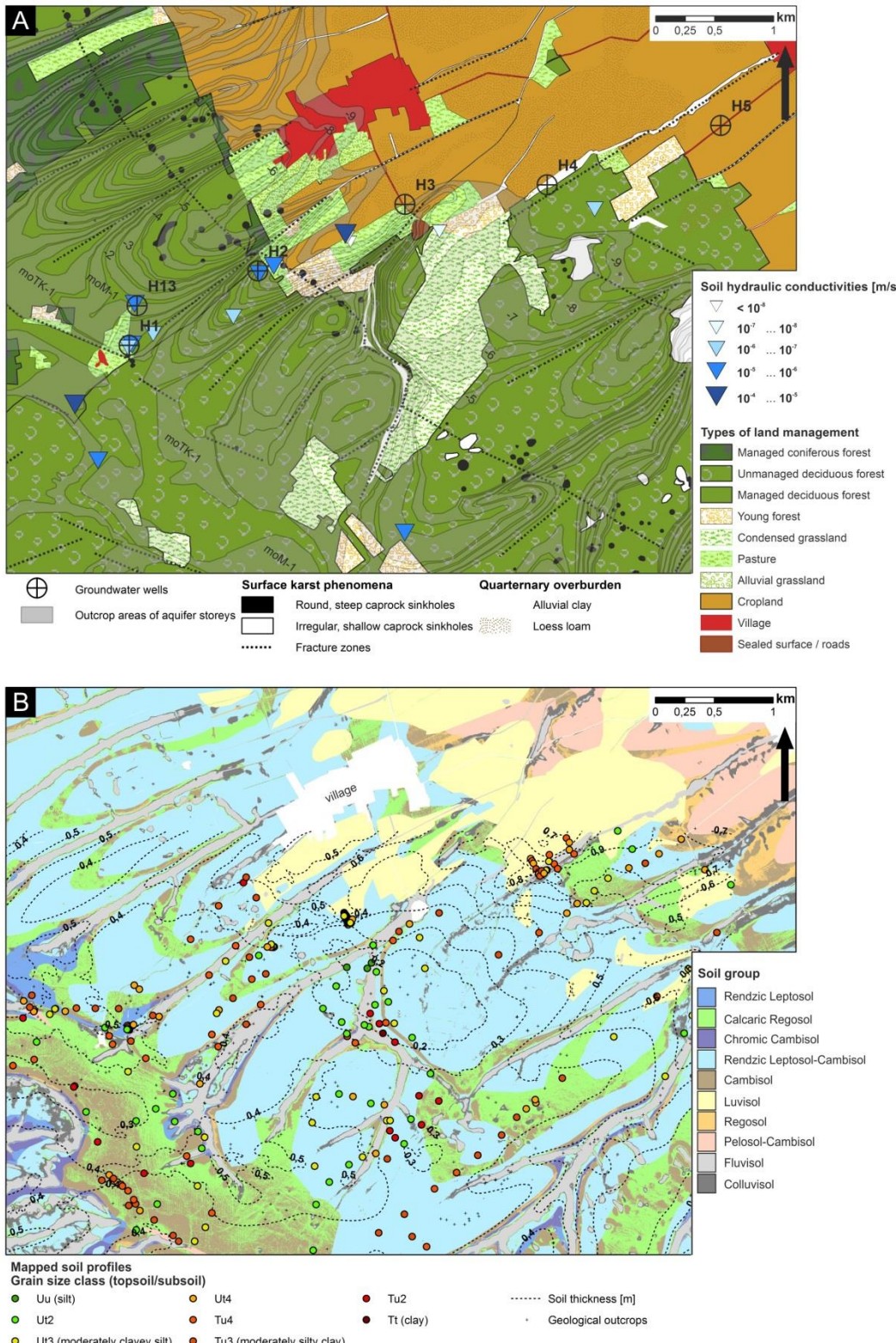

**Fig. 2.** (A): Types of land management, outcrop zones of aquifer storeys, sinkhole lineaments, potential fracture zones and
measurement points for soil hydraulic conductivities. Aquifer storeys which are lower in stratigraphy outcrop in higher
positions on the Hainich hillslope. The AquaDiva well transect H1 to H5 (also shown) covers hillslope regions from the
summit (H1) to the footslope (H5). (B) Conceptional soil map showing calculated soil groups and mapped calibration data
points. Signatures of the mapped soil profiles represent the grain size class (soil category) of the topsoil. Interpolated isolines
of mapped soil thickness show increasing thickness towards the NE and towards the transverse valleys.
**Fig. 3**

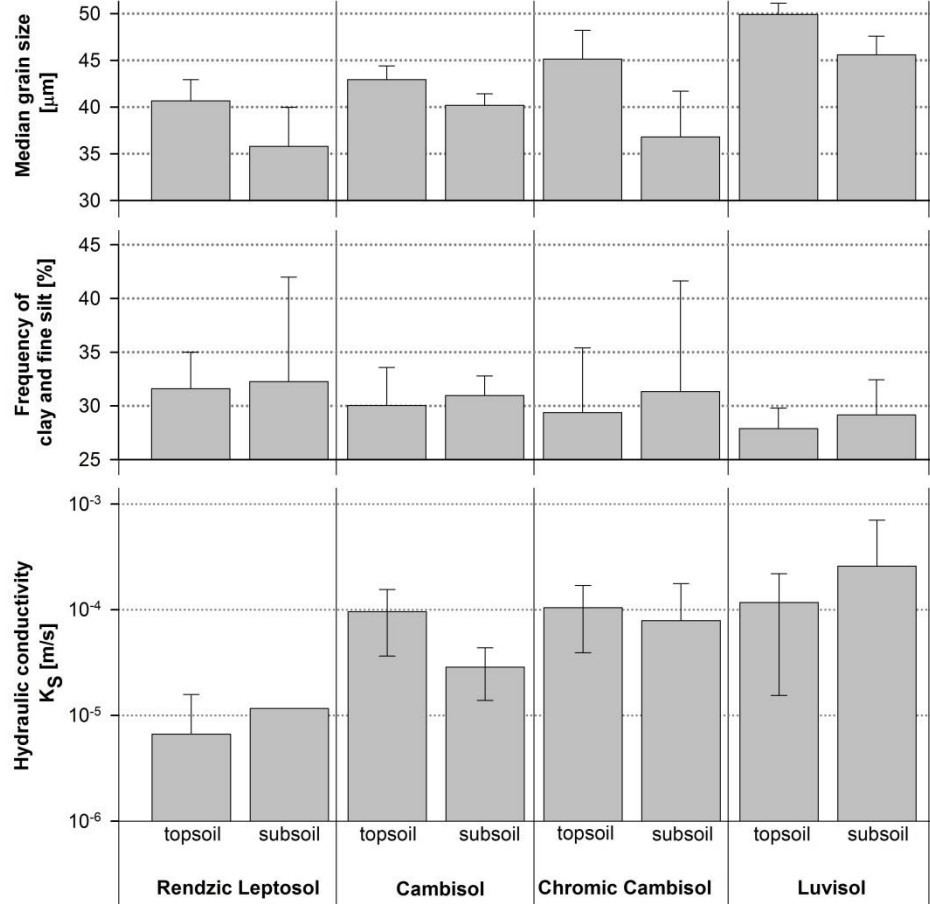

**Fig. 3.** Physical properties of topsoil/subsoils and parent rocks of the four major soil groups. (A): Median frequencies of the
fine grain size fractions (clay + fine silt) showing increasing proportions towards the bedrock, respectively. (B): Median soil
hydraulic conductivities of soil groups showing higher median soil hydraulic conductivity in Chromic Cambisols and
Luvisols compared to the Rendzic Leptosols and Cambisols and general decreases in hydraulic conductivity towards the
subsoil. The error bars describe the root of the estimation variance of the average.
**Fig. 4**

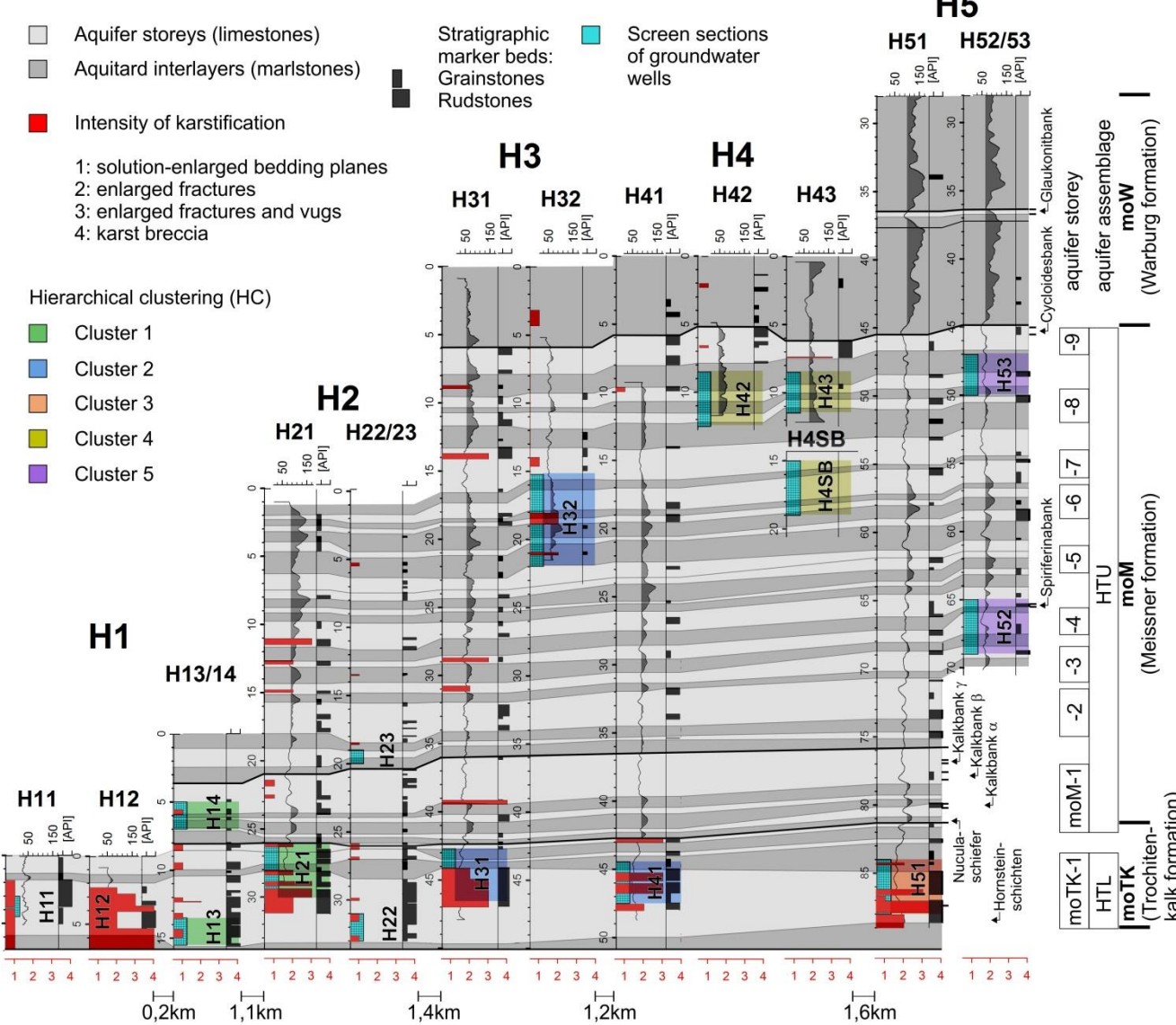

**Fig. 4.** Graphical correlation of marlstone/claystone intervals in gamma-ray logs, biostratigraphic limestone marker beds
(grainstones/rudstones) and the degree of karstification (red bars and red scale bar below: solution-enlarged bedding planes
to karst breccia). The geological aquifer correlation is cross-checked with the hierarchical clustering of hydrochemical
parameters.
**Fig. 5**

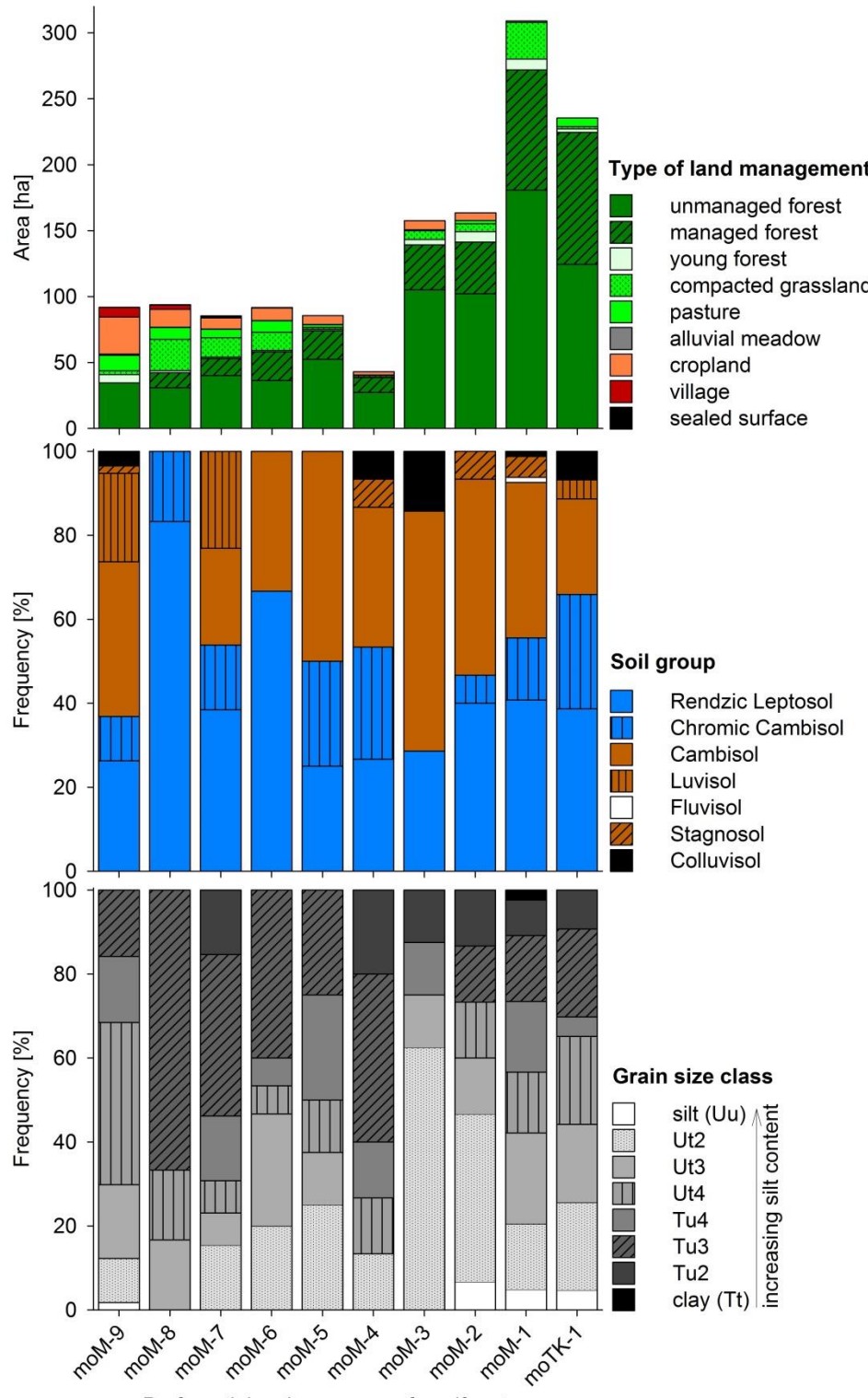

**Fig. 5.** Surface and subsurface properties of the aquifer outcrop zones (moTK-1 to moM-9) on the Hainich hillslope. Area
sizes are related to the 29 km² area of this study. (A): Absolute abundances of land management types in the preferential
recharge areas of the aquifer storeys. The two basal aquifer storeys are characterized by the largest aquifer outcrop areas and
the highest amounts of forest within these areas. Agricultural land management increases towards the higher aquifer storeys
(moM-2 to moM-9). (B): Relative abundances of soil groups within the aquifer outcrop zones showing high diversitiy in all
aquifer outcrop zones depending on slope positions and quarternary loess loam/alluvial clay coverage, (C): Grain size groups
of the topsoils and subsoils in the aquifer outcrop zones.
**Fig. 6**
**Fig. 6.** Stratigraphic succession of the Upper Muschelkalk with stratigraphic marker beds (left), a gamma-ray log, aquifer
assemblages HTL/HTU, aquifer storeys moTK-1 to moM-9 and the average chemical compositions of monitored
groundwater (left: ions of carbonate/sulphate minerals; middle: redox-sensitive ions; right: ions which are potentially related
to the type of land management). The color code of hierarchical clusters is identical to Fig. 4.
**Fig. 7**

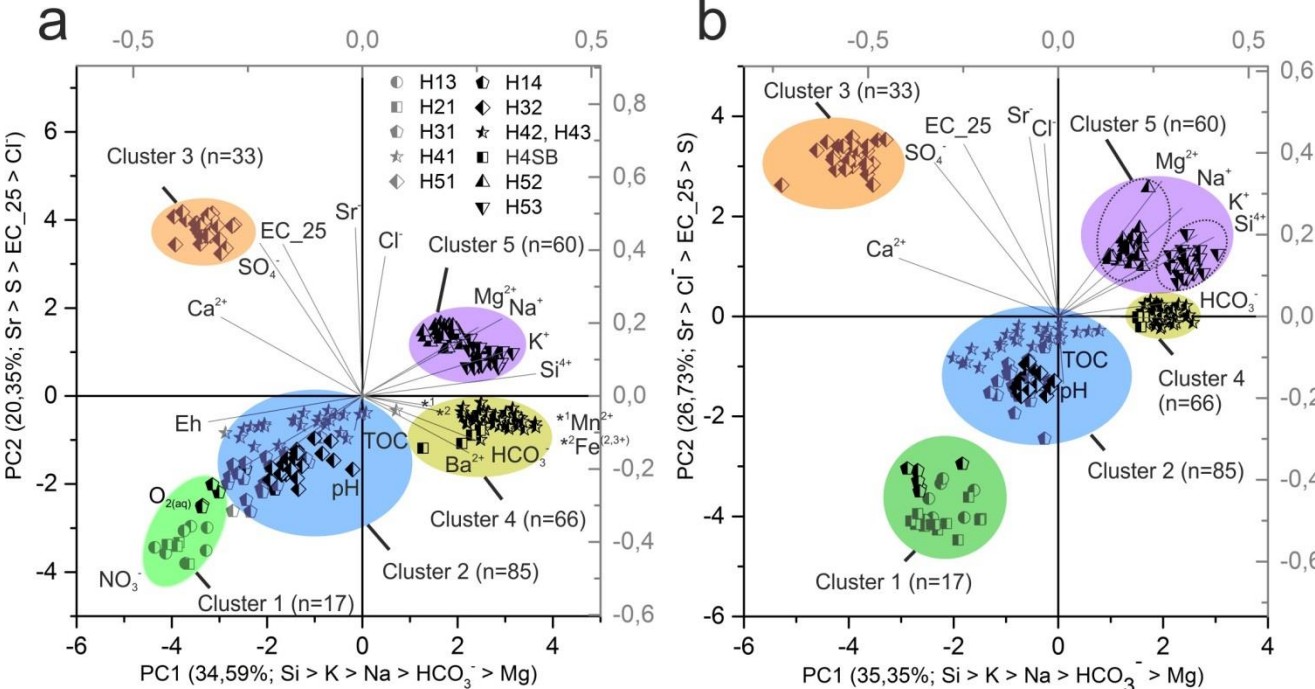

**Fig. 7.** Principal component analysis (PCA) biplot for the complete parameter set (a) and for the limited parameter set
(b) without redox related parameters. Five clusters can be distinguished in both parameter sets. Samples within the clusters
are identical for both PCA plots. Factor loads for PC1 and PC2 are displayed as labels of the x/y-axis as well as the sequence
of factor loads for individual components. The color code of hierarchical clusters is identical to Fig. 4.
**Fig. 8**

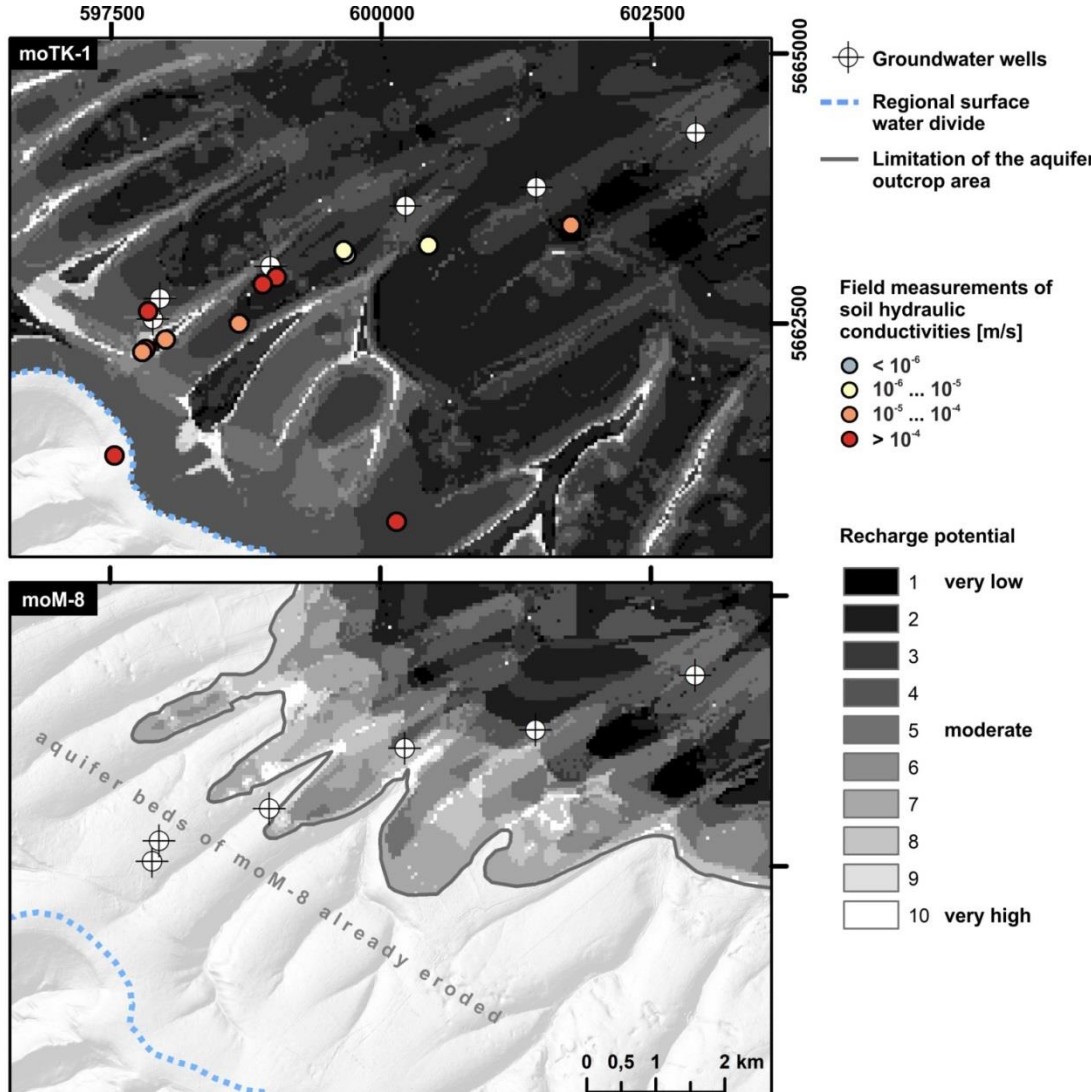

**Fig. 8.** Recharge potential maps for the two contrasting aquifer storeys moTK-1 (upper map) and moM-8 (lower map). The
recharge potential is a qualitative indicator for infiltration and percolation towards selected aquifer storeys. The brighter the
output color, the higher the recharge potential. Aquifer outcrop zones show highest recharge potentials, followed by sinkhole
lineaments and (NE-SW orientated) transverse valleys. Recharge potential decreases drastically with increasing thickness of
overburden strata towards the NE. Datapoints of field measurements are displayed in the left map (moTK-1 recharge
potential) and color-coded with respect to their soil hydraulic conductivity; Data sources: DEM ©GeoBasisDE/TLVermGeo,
Gen.-Nr.: 7/2016.

**Fig. 9**

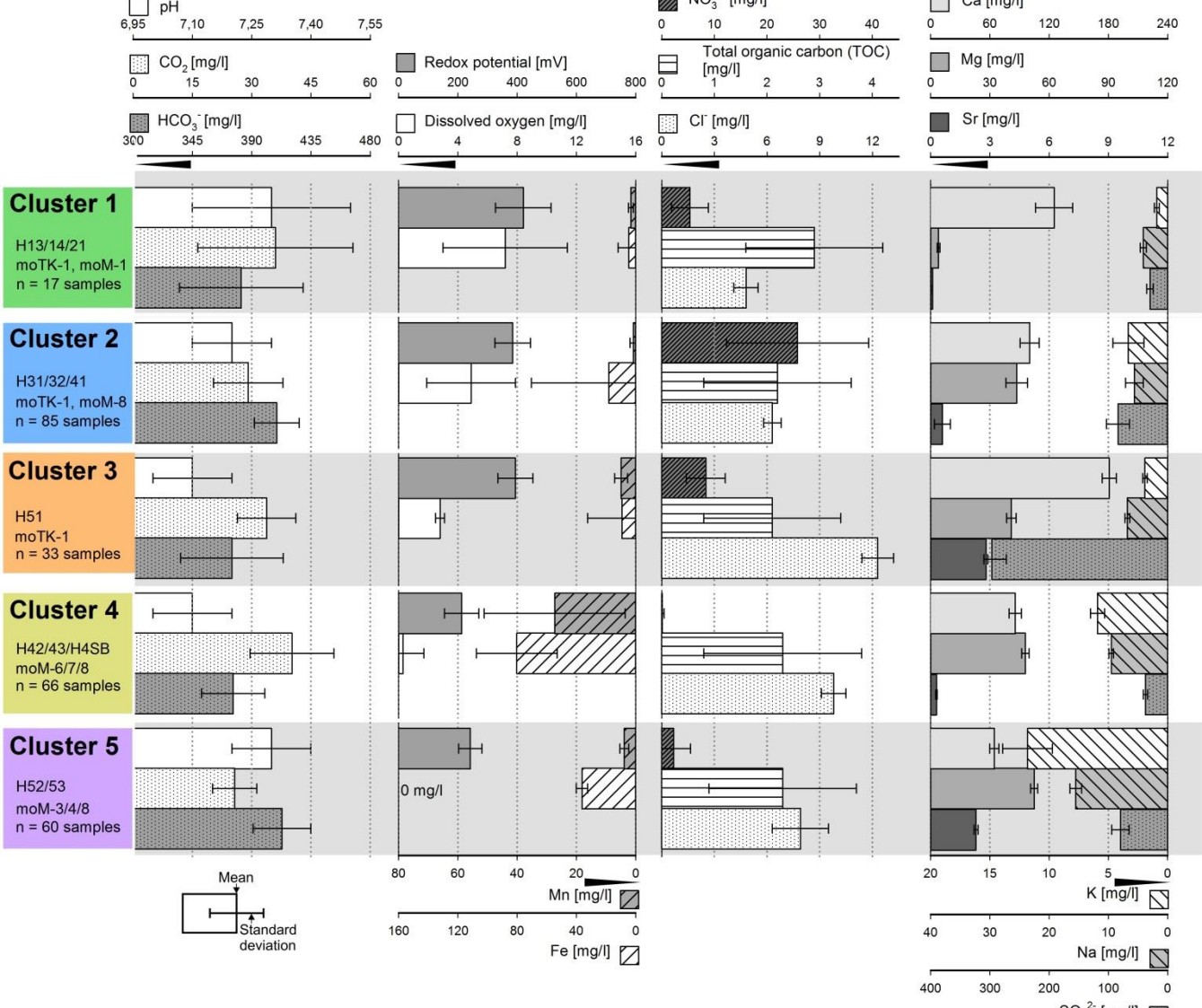

**Fig. 9.** Average groundwater chemistry of the hierarchical clusters 1-5 (left: groundwater wells and aquifer storeys within the clusters). First column: parameters related to the carbonate-$CO_2$-equilibrum, 2nd column: dissolved $O_2$, redox potential and redox-sensitive ions, 3rd column: ions which are potentially related to land management, 4th column: ions related to the dissolution of either carbonate/sulphate or clay minerals. The error bars describe the root of the estimation variance of the average. The colour code of hierarchical clusters is identical to Fig. 4.

**Fig. 10**

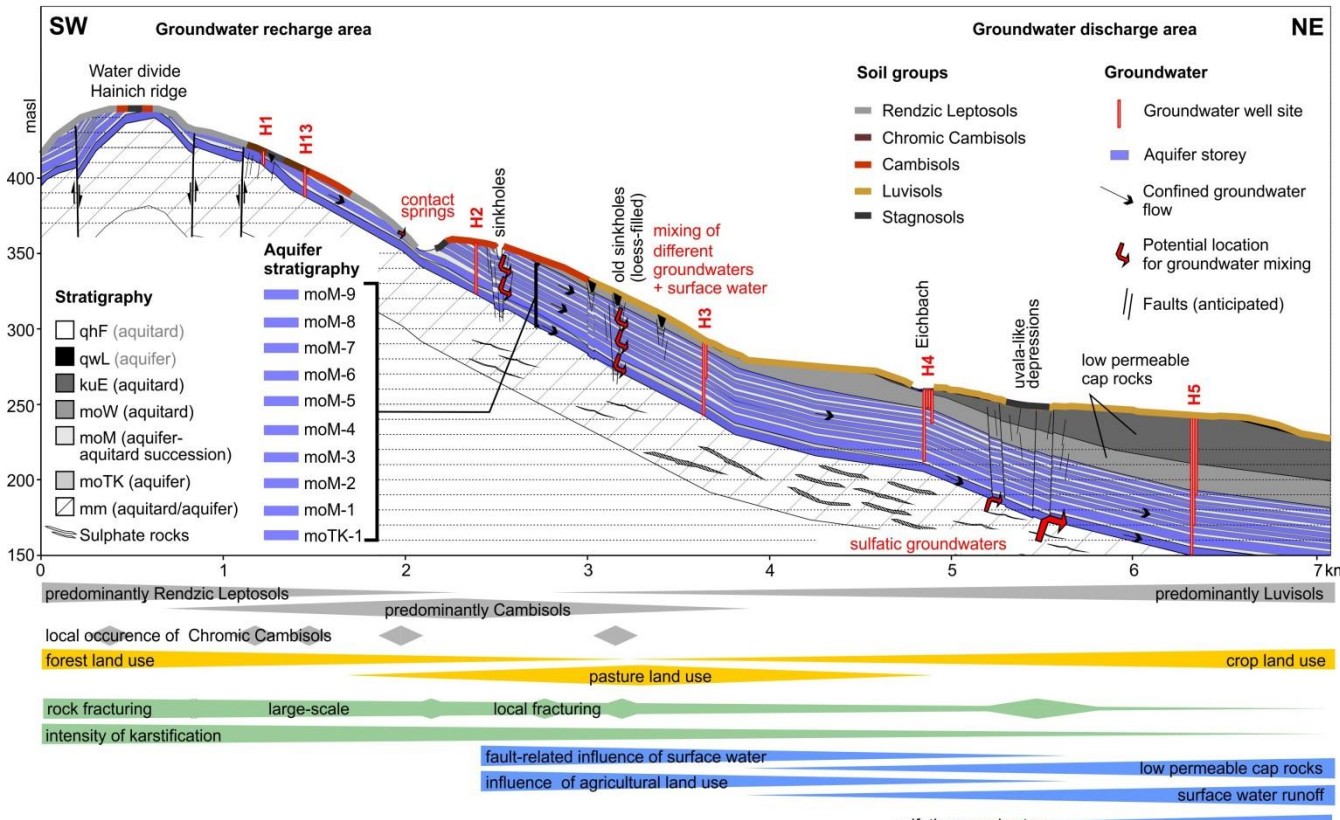

**Fig. 10.** Conceptual cross section of the Hainich hillslope showing the overall geological structure with the ten aquifer
storeys illustrating preferential aquifer recharge zones in the summit to midslope region as well as the discharge direction.
Lineaments of caprock sinkholes crossing the potential groundwater flowpath represent potential zones for descending or
ascending cross-formational flow and also the mixing of different types of groundwater.

