# Peer review of "Aquifer configuration and geostructural links control the groundwater quality in thin-bedded carbonate/siliciclastic alternations of the Hainich CZE, Central Germany"

_Hydrology and Earth System Sciences, 2016_

## Referee Comment (RC1) · Anonymous Referee #1 · 28 Oct 2016

Kohlhepp et al., 2016. Pedological and hydrogeological setting and subsurface flow structure of the carbonate-rock CZE Hainich in western Thuringia, Germany, Hydrol. Earth Syst. Sci. Discuss., doi:10.5194/hess-20160374, 2016

This paper reported a comprehensive investigation of subsurface carbonate-rock aquifer in western Germany, including stratigraphy, geochemistry, human factors, structure geology and karst heterogeneity. The authors did a lot of field works, collected many data and conducted a very nice statistic analysis. However, I think current manuscript still needs lots of works to be published on HESS, due to the reasons

as follow. I encourage the authors to address the scientific issues, restructure the manuscript and resubmit the paper.

1) Generally speaking, the structure of this manuscript is more like a hydrogeological survey report or groundwater resource summary, not a research article. Why did you do this study? What scientific questions are answered in this paper? The author listed three aims in the introduction part, but it seems the authors are trying to address so many issues in one manuscript, and bring difficulties for readers to follow up. The first aim is obviously not a science question but more like a geological background by the survey. The second and third aims are significantly different. Therefore, the detail demonstration of the connections between these two aims is highly expected. I actually suggest the authors to focus on one aim only in the paper. Also, it's very important to highlight the research purposes and the novelties in the title, abstract and conclusion parts.

2) Because it is a research article instead of report, the authors are expected to explain why Hainich CZE is important and interesting to study. Are there any special geological characteristics? I'm not familiar to the hydrogeological setting in Germany, but I assume that carbonate-rock structures are widely distributed. Is Hainich CZE a typical karst aquifer in Germany? All of those are necessary to be fully illustrated in the manuscript.

3) The authors used more than half of the words in this manuscript to introduce and describe the field works and data collections. Again, I would recommend the authors to focus on the discussion of statistic analysis (PCA and cluster analysis) of geochemistry data, and address the effect of karstification and hydrological stratigraphy on groundwater quality/hydrogeocemistry (section 4.2).

4) The authors mentioned the effects of fault zones on groundwater chemistry with dissolution-enlarged fractures. Hydraulic conductivity through the faults in karst aquifer can be larger in several magnitudes, due to the dissolution of carbonate-rock dissolution. Does dissolution play a more important role rather than faults? More explanations

are expected. On the other hand, dual-permeability hydrological characteristics are commonly observed in karst aquifers. The authors should address some literature citations of flow properties in karst aquifer in the introduction. 5) The authors might not have enough data, especially the historical data before the beginning of sampling. But it is interesting to see any trends of geochemistry data variation along time, with changes of land use type and anthropogenic factors. And a discussion of the effect of contamination/pollution/human factors to data is desired.

6) In the end of section 4.2, the authors classify three modes of subsurface water flow in the karst aquifer. I would say the lineaments of sinkholes are not necessarily due to flow through open faults. Is there a possibility that bedding parallel in either unconfined and confined aquifer can cause lineaments of sinkholes as well? Probably just track the faults/fracture zones from geological map/structure survey.

7) Discussion 4.3 has weak relevant to the statistics analysis result. I don't think the authors have enough data to discuss karstification dissolution, so I recommend removing it.

8) To be honest, I didn't get the key points in the conclusion part. The authors do not need to mention the results of mapping and survey in the conclusion part. I suggest the authors to summarize the results of data analysis and emphasize the relationships. It might be better to make the statements by bullet points.

---

## Short Comment (SC1) · 22 Nov 2016

[supplement omitted: unrelated document]

---

## Referee Comment (RC2) · Anonymous Referee #1 · 20 Jan 2017

I agree with the clarification of scientific questions summarized by the authors in the reply. However, generally it's not easy and impossible to answer multiple questions in one research article, which makes the paper read messy. I would suggest the authors to address those questions (especially the vulnerability studies and the impacts of anthropogenic hazards) and reorganized the manuscript. The methods used in this paper need to be highlighted and explained explicitly to let the readers understand the work you did. And it's probably better to divide the method and site description as two sections in the manuscript.

The authors also mentioned the effects of faults with respect to the tectonic and exhumantion history, which is believed as an important topic in this paper. I encourage the authors to address it more.

A revised manuscript is necessary for further review.

---

## Author Comment (AC1) · 2 Feb 2017

We see the point made by the reviewer, that the presentation of the multiple aspects makes the paper hard to follow. We thus decided to focus on the following key question(s):

- How relates the intrinsic vulnerability of this widespread, yet scarcely addressed type of carbonate-rock settings (thin-bedded carbonate/siliciclastic alternations) to surface conditions (soil cover, land management, surface karst phenomena) and subsurface factors (bedding, aquitard-aquifer sequences, faulting, and subsurface karst)?

- Does this relation reflect in the hydrochemical state as a function of depth and slope positions?

In line with reviewer's suggestions, we will lay a strong focus on the intrinsic vulnerability by exemplifying how the surface and subsurface compartments interact with the fluids and affect hydrochemistry of the aquifers within such a rural catchment with only little anthropogenic impact.

The manuscript will be harmonized according to the suggestions of RC2:

- We will condense the characterization of the site in one chapter. The section on methods and site description will be divided in two compact sections.

- We will highlight the significance of the setting-specific site characteristics for the understanding of groundwater quality and subsurface architecture.

- We will emphasize anthropogenic hazards for an extended discussion of our results about intrinsic vulnerability.

- We will highlight the role of fault zones and caprock sinkholes for fluid flow based on additional results from mapping and digital elevation model (DEM) analysis.

Accordingly, we propose to revise the manuscript as follows:

- The uniqueness and capacities of the Hainich CZE will be illustrated in the introduction.

- We will highlight the importance of thin bedded carbonate-siliciclastic aquifer systems and the characteristics of these setting; this includes the description of flow paths which are predominantly fractures and subordinately karst cavities.

- We will extend the discussion of groundwater quality by emphasizing recharge-area surface properties and the subsurface structure. This will also include a discussion of "natural" and "anthropogenic" substances in the groundwater.

- We will emphasize the relationships between surface properties (land use, soil cover), karstification and aquifer stratigraphy for resulting groundwater hydrochemistry (following hess-2016-374-RC1).

- We will discuss intrinsic aquifer vulnerability.

- Fault zone inventory will be added to the results. Fault-related cross-formational flow (apart from fault zones) between the identified multi-storey aquifers will be discussed; this will also include a definition and explanation of caprock sinkholes.

- Although, groundwater quality fluctuations are very complex and will therefore be discussed in a separate research article, we will add an example of time series data (i.e. nitrate fluctuation) to the results section; this will help to assess mean values of hydrochemistry in the context of seasonal changes.

- Chapter 4.3 which is not well related to the hydrochemical clustering will be transformed to a much shorter discussion of the distribution of karstification subsequent to the hydrochemical discussion.

- The conclusions will be more focused on the relationship between land use, infiltration potential and hydrochemistry and the consequences for the "intrinsic vulnerability"

Please also note the supplement to this comment:
http://www.hydrol-earth-syst-sci-discuss.net/hess-2016-374/hess-2016-374-AC1-supplement.pdf

---

## Referee Comment (RC3) · E. Zehe (Referee) · 20 Jun 2017

Summary: I read this study with great interest as it is one of the rare efforts to characterize the entire critical zone in a holistic manner with respect to its hydraulic, (hydro-) geological and hydro-chemical characteristics. To this end the authors draw from a comprehensive and divers data set that has been collected in a rather complex and karstified hydrogeological setting in Thuringia. Specifically the combine a survey of soils and landuse, various geophysical techniques and drillings to infer on subsurface strata with data from a large amount of ground water wells and comprehensive hydro chemical data collected during several sampling campaigns. The authors synthesize the different data sources into a conceptional model of their study area through a combination of process based reasoning, expert knowledge and multivariate statistical methods.

Evaluation: The study is based on a very sound data set, the manuscript reads well and the joint discussion of the different data and their synthesis to conceptional model is appealing. But as it stands it remains a very sound and thorough study of a single case, because the authors miss quite obvious opportunities to address more generic questions. Furthermore, the characterization appears not so holistic. With respect to the treatment of the soil, the assessment steps barely beyond a soil standard survey. I thus encourage the authors to extend their analysis by addressing more generic questions using the beautiful data they have at hand. I hope the authors will find the following points helpful to further optimize their study for the reader and to fully explore the potential of their effort.

Major points:

- An obvious question that could be addressed is how much of the proposed experimental and monitoring effort is needed to come up with such a comprehensive assessment – or the other way around how much of the information can be left out before without changing the quality of the conceptual model. This question could be easily addressed by taking the presented insights as the best guess of the unknown truth and stepwise leaving out increasing amounts of for instance their hydro chemical data (in space and or in time) and perform the same multivariate analysis.

- The characterization of the soil is compared to the characterization of the deeper subsurface rather descriptive and follows mainly standard mapping approaches. I wonder whether any data on permeability and infiltrability where collected? Even if, not the analysis of the available soil types lacks behind its potential to infer on recharge areas. The latter depend on the ks, retention properties and apparent preferential pathways

and the spatial pattern thereof. Even if the latter were not mapped, one could use a pedo transfer function to estimate ks and compile a geostatistical analysis with interpolation or even conditional simulation. This would yield an estimate on potential hot spots for recharge.

Technical details:

- Generally the figure caption are very brief. Figure 4: how much variance is explained by the first two principle components? I didn't find it in the text, would be nice to provide it here.

- I guess the color code in Figure 5 represent the cluster in Figure 4. Would be helpful to add that to the caption, also of Figure 6.

- Figure 7. Is the error bar the standard deviation of the sample or the root of the estimation variance of the average. I guess the latter is more appropriate to infer on significant differences.

- The reference in the section 4.1 (infiltration properties) is not the most recent one, particularly not with respect to preferential flow. Furthermore, this part is rather descriptive and could possibly be written without the data you have.

Best regards,

Erwin Zehe
* * *

---

## Author Response (AR1)

**Author´s response (HESS-2016-374)**

**1. Point-by-point response to review 1**

**1) Generally speaking, the structure of this manuscript is more like a hydrogeological survey report or groundwater resource summary, not a research article. Why did you do this study?**

We are sorry that the reviewer got this impression but disagree with the reviewers rating of the MS. Indeed, the paper presents an in-depth and thorough analysis of the subsurface. This requires among others also a hydrogeological survey. However, we claim that this is by far not enough to reach the goals of our study:

- Understand the hydraulic and biogeochemical functioning of the subsurface in a so far not adequately addressed setting (thin-bedded carbonate-siliciclastic aquifer bedrock) provided by the Hainich CZE.
- Explore the links and feedbacks between surface and subsurface.
- Demonstrate that a holistic approach that considers the surface and subsurface factors is indispensable for the mechanistic understanding of the coupling of surface and subsurface compartments, fluid dynamics, biogeochemical element cycling and ecology in the critical zone.

The extent to which we have analyzed the subsurface goes far beyond a classical hydrogeological survey for two major reasons:

(1) We consider and reconstruct all subsurface compartments, i.e., the soils sensu strictu, the unsaturated and the saturated bedrock in depth and combine it with the actual type of land management.

(2) We present and put in action a novel and comprehensive multi-method approach that combines geological, hydrogeological, hydrochemical, pedological, structural geological, mineralogical, and geophysical tools and methods to assess the effective hydraulic and transport structures of the subsurface compartment and to explain the state and evolution of the groundwater chemistry.

With this multi-method approach, we provide

- a detailed aquifer stratigraphy for understanding horizontal and vertical connectivity
- a geological and pedological mapping of aquifer outcrop areas for the localization of preferential infiltration zones, as vertical infiltration is highly minimized by argillaceous confining beds
- a determination of soil hydraulic conductivities and grain size distributions/soil texture properties
- a mapping of the types of land management as well as of relief landforms/slope positions
- the analysis of groundwater quality in different slope positions and aquifer storeys by means of multivariate statistics

We further stress the point that – based on our multi-method approach - the actual spatial distribution pattern of the hydrochemistry observed in the different groundwater "clusters" is controlled by the residence time and the interactions of both (I) the infiltrating precipitation and seepage with the interfaces in soils and (II) unsaturated and saturated bedrock formations.We are convinced that this novel, comprehensive and synoptic approach is indispensable for interpreting groundwater quality in essentially all groundwater systems and that without this multi-method-approach these quite distinct and synoptic results would not have been reached. Although there are a small number of published examples characterizing aquifer vulnerability/human impact, none of these studies characterizes pristine groundwater quality in comparison with aquifer stratigraphy, soil hydraulic properties and land management in a hillslope groundwater catchment. We revised the manuscript to make theses aims clearer to the reader.

**1. Response to review 1**

**What scientific questions are answered in this paper?**

The guiding scientific question is: What are the essential and necessary information to be gathered in order to understand the factors that control the hydraulic and biogeochemical functioning of the subsurface and to explore the links and feedbacks between surface and subsurface as a function of land use. An additional aspect we aimed for to address is the specific setting provided by the Hainich CZE. It represents a widely distributed, yet scarcely described and in previous publications not adequately addressed setting of thin-bedded mixed carbonate-siliciclastic strata in hillslope terrains. Theses settings provide groundwater reservoirs with importance as drinking water supply and a significant geographical distribution not only in Europe (e g. "Germanic Triassic", Khuff formation (Middle East)).

**The author listed three aims in the introduction part, but it seems the authors are trying to address so many issues in one manuscript, and bring difficulties for readers to follow up. The first aim is obviously not a science question but more like a geological background by the survey. The second and third aims are significantly different. Also, it's very important to highlight the research purposes and the novelties in the title, abstract and conclusion parts. Therefore, the detail demonstration of the connections between these two aims is highly expected. I actually suggest the authors to focus on one aim only in the paper.**

Once again, we are sorry that the reviewer got this impression. We tried to counteract that by rearranging the manuscript to clarify why we approached the scientific questions in that way. Of course, the multi-method-approach has many aspects. In the revised paper we emphasized the need and explained in more detail the relations and links of the different aspects in view of the aims of the MS. For example, we present the relationships between surface and subsurface influences on hydrogeochemistry in the introduction, discussion and conclusion chapters. We substantiated with our dataset how close groundwater quality is linked to the recharge area characteristics.

We also clarified the goals and research questions in the introduction. The paragraph now reads (page 3, line 28 ff.):
*The goals of our study are to (1) understand the hydrogeological and biogeochemical functioning of the subsurface in a so far not adequately addressed setting provided by the Hainich CZE, (2) explore the links and feedbacks between surface and subsurface, and (3) to demonstrate that a holistic multi-method-approach, that considers the surface and subsurface factors, is indispensable for the mechanistic understanding of the coupling of surface and subsurface compartments, fluid dynamics, biogeochemical element cycling and ecology in the critical zone.*
*To reach these goals, the following research questions are answered:*
*(I) How is the critical zone comprised and connected in the hillslope setting of a thin-bedded, mixed carbonate/siliciclastic succession? (II) How do spatial arrangement, particularly outcrop patterns and geostructural links (karst features like caprock sinkhole lineaments) impact compartment connection, (intrinsic vulnerability) and groundwater quality? Do groundwater quality in contrasting summit/midslope and footslope wells reflects surface influences?" And finally: (III) what are the main control parameters for groundwater quality? What are the reasons for particular hydrogeochemical conditions within the multi-storey/hillslope aquifer system?*

**2) Because it is a research article instead of report, the authors are expected to explain why Hainich CZE is important and interesting to study. Are there any special geological characteristics? I'm not familiar to the hydrogeological setting in Germany, but I assume that carbonate-rock structures are widely distributed. Is Hainich CZE a typical karst aquifer in Germany? All of those are necessary to be fully illustrated in the manuscript.**

We agree and have changed the MS accordingly. We introduced a section on the Hainich CZE within the introduction which illustrates its importance and relevance to the greater scientific community. This section now reads as follows (page 4, line 23 ff.):

*The Hainich CZE is a multifarious environmental laboratory for multiscale geo- and bioscience research, as*

*(1) both, the so scarcely described geological setting (alternations of marine thin-bedded limestones and marlstones) and the hillslope relief/sloping aquifer configuration are common and widely distributed*

*(2) the monitoring plot and well transect (see Küsel et al., 2016) provides a unique access to the multi-layered aquifer system of the common setting*

*(3) it represents a rare anthropogenically low-impacted (non-contaminated) cultural region in central Europe with a very extensive type of land management during the last centuries, allowing the investigation of natural (surface) signal transformation in the pristine aquifer systems and of ecosystem functioning*

*(4) the geostructural and lithological properties were found to be predictable and thus enable the tracking of biogeochemically cycling and quality development within single aquifer storeys*

*(5) the Hainich it is a regionally important groundwater recharge area in Thuringia (Fig. 1) and an example of peripheral groundwater supply for water deficient sedimentary basins (Rau and Unger, 1997; Hiekel, 2004)*

We have exemplified the details of the geological setting, which is widespread and common, but scarcely described in literature, in more detail. We added important information on the geological characteristics. The paragraph now reads as follows (page 4, line 2 ff.):

*The Hainich Critical Zone Exploratory covers 430 km² of a hillslope subcatchment of the Unstrut river in northwestern Thuringia, Central Germany (Fig. 1). It is bounded by the recipient Unstrut river and the distribution area of the Upper Muschelkalk formations. The Hainich represents a NW-SE orientated geological anticline that is developed topographically as a low mountain range with a steep western, and a moderately inclined eastern flank (Jordan and Weder, 1995). The study area is located at the eastern flank of the Hainich hillslope, thtat shows a geostructural buildup with NE dipping strata in the direction of the syncline of Mühlhausen-Bad Langensalza (Kaiser, 1905; König, 1930; Patzelt, 1998; Wätzel, 2007). A tectonical uplift, faulting and tilting of strata is assumed for the Late Cretaceous in analogy to the surrounding horst structures Thüringer Wald (in the S) and Harz (in the N; compare to Voigt et al., 2004; Kley and Voigt, 2008). The outcropping strata in the study area comprise sedimentary rocks (Middle/Upper Muschelkalk and Lower Keuper - subgroups of the Middle Triassic). According to the German stratigraphy (Deutsche Stratigraphische Kommission, 2002), the Diemel formation (Middle Muschelkalk), Trochitenkalk, Meissner, and Warburg formation (Upper Muschelkalk) and the Erfurt formation (Lower Keuper) outcrop in the study area. The Upper Muschelkalk subgroup, which hosts the target aquifers of the Hainich CZE, is organized by bio- and lithostratigraphic marker beds (Ockert and Rein, 2000; Kostic and Aigner, 2004). Previous studies hydrostratigraphically organize the Upper Muschelkalk subgroups into a Hainich Transect Lower Aquifer Assemblage (HTL) and a Hainich Transect Upper Aquifer Assemblage (HTU; Küsel et al., 2016).The area belongs to the Cfb climate region (C: warm temperate, f: fully humid, b: warm summer) according to the Köppen-Geiger classification (Kottek et al., 2006) and exhibits a leeward decline in areal precipitation and increasing mean air temperature*

*from the Hainich ridge (> 900 mm/y; 7.5-8 °C) to the Unstrut valley (< 600 mm/y; 9-9.5 °C; long term average 1970-2010, TLUG, 2016). The intensively investigated study area is limited to a 29 km²-subarea of the Hainich CZE, that surrounds the soil and groundwater monitoring transect (Küsel et al., 2016).*

**3) The authors used more than half of the words in this manuscript to introduce and describe the field works and data collections. Again, I would recommend the authors to focus on the discussion of statistical analysis (PCA and cluster analysis) of geochemistry data, and address the effect of karstification and hydrological stratigraphy on groundwater quality/hydrogeochemistry (section 4.2).**

As we introduced a new and contrasting aquifer stratigraphy of a common, but barely addressed setting, we decided for a comprehensive method and results presentation. Nevertheless, we summarized and abbreviated the site description and methods chapter (together 2742 words) and extended the results (2725 words) and discussion (4030 words). We agree that a focused discussion of groundwater quality is necessary. Thus, we extended the discussion of geochemical data in the discussion:

- In chapter 5.1.2 (page 12, line 14 ff.), groundwater chemistry was discussed in the context of the aquifer configuration and groundwater development.
- Groundwater flow directions (5.2.5, page 15, line 5 ff.) are interpreted by means of hydrogeochemistry, karst phenomena and the knowledge of local geology.
- Finally the provenance of groundwater from different hierarchical clusters was interpreted (5.3, page 15, line 40 ff.).

**4) The authors mentioned the effects of fault zones on groundwater chemistry with dissolution-enlarged fractures. Hydraulic conductivity through the faults in karst aquifer can be larger in several magnitudes, due to the dissolution of carbonate-rock dissolution. Does dissolution play a more important role rather than faults? More explanations are expected.**

The study area is characterized by old NW-SE orientated fracture systems that are described in regional literature as preferential flow paths. This fracture system is now described in the following text passages:

*Second order relief elements are also the two NW-SE oriented lineaments of more than eighty caprock sinkholes (which are passive karst phenomena) with up to 80 m in diameter (page 7, line 30 ff.).*

*Two contact springs in the recharge area (Grauröder Quelle, Ihlefeldquelle) and two karst springs in the discharge area (Kainspring, Melchiorbrunnen, coupled to NW-SE oriented fault zones, ...(page 10, line 26 ff.).*

*As caprock sinkholes are arranged in lineaments (Fig. 2 A) parallel to regionally known fault orientations, it is very likely that they are coupled to the penetration of surface/subsurface water at fracture zones (compare to: Smart and Hobbs, 1986; Worthington, 1999; Klimchouk, 2005), that promote preferential recharge (Smart and Hobbs, 1986; Suschka, 2007) (page 13, line 20 ff.).*

Our descriptions of dissolution-enlarged fractures are limited to our drill cores material which had been recovered from wells which had not been drilled directly into fault zones. Our assumption that fractured/more permeable rocks are predisposed to a stronger karstification, resulting in a greater overall permeability in fault/fracture zones was added to the discussion chapter 5.2.5 and reads as follows (page 15, line 19 ff.):

*Zones with enhanced fracture indices (fractures/m drill core) in well sites H3 as well as consistent Fe/Mn-oxide fracture*

*walls in all aquifer storeys of this site points to a quick and cross-formational descending flow of oxygenated groundwater (thus with low Fe/Mn mobility; Hem, 1985) via vertical master joints (term: Dreybrodt, 1988; Ford and Williams, 2007) close to potential fracture zones. This is supported by higher concentrations in Na+, K+, NO3-, Cl- and TOC in the deeper site of H3 compared to the shallower well of this site, related to agriculture and fertilizing (Matthess, 1994; Kunkel et al., 2004). Cross-formational descending flow in shattered or fractured rocks is assumed to take place in fracture zones (Worthington, 1999; Goldscheider & Drew, 2007), tracked by lineaments of caprock sinkholes (Mempel, 1939; Hoppe, 1962; Smart and Hobbs, 1986; Jordan and Weder, 1995).*

Fracture zone-related cross-formational flow is also discussed in chapter 5.3 (page 16, line 16):

*Oxygenated groundwater in both aquifer storeys (thus with low Fe/Mn mobility; Hem, 1985) and a high degree in rock fracturing and fracture mineralization with Fe-/Mn-oxide minerals (aquifer storeys moM-9/8/7 and 6), point to a vertical penetration with near surface groundwater via fracture zones through all aquifer storeys. Enhanced concentrations in $Na^+$, $K^+$, $NO_3^-$, $Cl^-$ and TOC (Fig. 7) are likely related to agriculture and fertilizing (Matthess, 1994; Kunkel et al., 2004) around the sinkhole lineaments. As nitrate, which is generally derived from agricultural fertilizers (Agrawal et al., 1999; Jeong, 2001), is still present in the deep aquifer waters of site H3, vertical bypassing through master joints (term: Dreybrodt, 1988; Ford and Williams, 2007), must be faster than the denitrification process. Quick infiltration is here mostly related to the preferential sinkhole recharge, as the soils (Luvisol/Cambisol, both with low-conductive subsoils and a high degree in lateral soil interflow) in the outcrop zones bear only moderate to poor soil hydraulic conductivities. In general groundwater in the discharge of exceptionally highly conductive preferential recharge spots can reflect surface signals, even in stratified aquifer/aquitard successions. These zones are therefore suggested to be of uppermost importance for groundwater protection.*

In our thin-bedded mixed carbonate-siliciclastic setting, karstification effects are not very distinct (in comparison with classical karst/unconfined karst). Karst/dissolution phenomena are limited to formations with very high limestone-marlstone ratio (here: Trochitenkalk-formation). Other parts of the stratigraphic succession may be regarded more or less as fracture aquifers. We highlighted this important information in the results. The paragraphs of chapter 4.3.1 and 4.3.2 are formulated as follows:

*The 7 m thick Trochitenkalk formation (moTK) with thick (5 30 cm), gray, coarse bioclastic limestones (mainly rudstones with the rock-forming fossil Encrinus liliformis) forms a carbonate-rock-fracture aquifer with minor karstification (intrastratal karst according to Ford and Williams, 2007) (page 9, line 11 ff.).*

*Limestones in the Meissner formation are fracture aquifers and their rock matrices are predominantly composed of calcite and trace amounts of dolomite, quartz, illite and feldspar (page 9, line 25 ff.).*

*Flow paths are predominantly fractures and matrix porosity is lower than 5 % in all stratigraphic intervals. Although it is of karst-fracture type with partially solution-enlarged fractures, the karstification and the development of conduits is limited and concentrated at the formation´s very base (page 9, line 42 ff.).*

*The Meissner formation (moM) contains limestone-fracture aquifers which are interbedded marlstone-aquitards on the decimeter to meter scale. Limestones of this formation are almost exclusively fracture aquifers with very little matrix porosity, concentrated at certain thickly bedded limestone marker beds (page 10, line 5 ff.).*

**On the other hand, dual-permeability hydrological characteristics are commonly observed in karst aquifers. The authors should address some literature citations of flow properties in karst aquifer in the introduction.**

We agree with the reviewer that dual-permeability flowpaths are typical for classical/massive karst sites. Obviously we failed to distinguish our thin-bedded carbonate-karst setting from classical karst sites. Thus, we highlighted these differences in the manuscript title, abstract and introduction.

*Aquifer configuration and geostructural links control the groundwater quality in thin-bedded carbonate/siliciclastic alternations of the Hainich CZE, Central Germany* (manuscript title)

*This CZE represents a widely distributed, yet scarcely described setting of thin-bedded mixed carbonate-siliciclastic strata in hillslope terrains* (page 2, line 8 ff.).

*Here, the bedrock of fractured, mixed carbonate-/siliciclastic alternations represents a widely distributed, yet scarcely described geological setting* (page 3, line 26 ff.).

Dissolution-related conduits are very limited and matrix porosities are very low (5 %). We added this information to the results (chapter 4.3.2):

*Flow paths are predominantly fractures and matrix porosity is lower than 5 % in all stratigraphic intervals. Although it is of karst-fracture type with partially solution-enlarged fractures, the karstification and the development of conduits is limited and concentrated at the formation´s very base* (page 9, line 42).

As groundwater flow properties are not in the scope of our study, we changed the title of the manuscript to:
*Aquifer configuration and geostructural links…* (manuscript title)

We do not agree with the reviewer, that literature citations of flow properties are necessary in the introduction, as it is not on-topic to the following aspects discussed within the introduction:

- Near surface groundwater quality is strongly related to the percolation of water through soils, the unsaturated zone and the aquifers Groundwater quality is hereby mainly controlled by bio/geochemical fluid-rock/soil interactions in these compartments (compare to page 3, line 2 ff.).
- A holistic investigation of both surface and subsurface is necessary for understanding groundwater quality. Although there are comprehensive studies published (i.e. Gleeson et al., 2009), these studies do not discuss pristine groundwater quality (compare to page 3, line 15 ff.).
- As a mixed carbonate-/siliciclastic setting, Hainich CZE represents a widely distributed, yet scarcely described geological setting (see page 3, line 26 ff.).

We thus described groundwater flow modes in the discussion. The paragraph now reads as follows (page 12, line 36 ff.):

*Within the aquifer-aquitard-sandwich, fast conduit groundwater flow, which is typical for karstified carbonate rocks (Wong et al., 2012), likely takes place in the moTK-1 and partially in the moM-1 aquifer storeys, whereas slow diffusion in slightly fractured, thin aquifers beds is anticipated in the moM-2 to moM-9 aquifer storeys. Confined diffuse flow is considered*

*laminar and takes place in interparticle pores and fractures of dense limestones with low primary porosity (Smart and Hobbs, 1986). This results in the well oxygenated moTK-1 and moM-1 groundwaters and a significant oxygen consumption/deficiency (coupled to the mobility of Fe2+ and Mn2+-ions and the low mobility of NO3- and SO4- ions) in the moM-2 to 9 groundwater, resulting in completely different milieu conditions for the biogeochemical processes and for the life in the subsurface as well.*

**5) The authors might not have enough data, especially the historical data before the beginning of sampling. But it is interesting to see any trends of geochemistry data variation along time, with changes of land use type and anthropogenic factors.**

As the fluctuations of groundwater levels and hydrogeochemical data is very dynamic and complicated (with respect to seasonality, rainstorm events and response times), we decided to present this issue in a second research article (in preparation). Nevertheless, we agree with the reviewer that a brief description of groundwater fluctuations is desirable and added a chapter (4.4.3) to the results (page 11, line 12 ff.). The chapter now reads:

*4.4.3 Fluctuations of groundwater levels and quality*
*Groundwater levels are confined in footslope wells. In the wells of site H5, groundwater rises 30 m (H53), 50 m (H52) and 70 m (H51) higher than the base of the screen section. Fluctuations of groundwater levels in our monitoring wells range from 1-3 m in the hilltop recharge area (well site H13) to more than 25 m in the groundwater transit area (well site H5). The groundwater level fluctuations show a strong seasonality with annual highstands (March to April) and lowstands (October to December). Monitoring wells of different sites and screen depths differ in average concentrations of the major solutes. These spatial differences are generally higher than the seasonal fluctuations in the wells. An exception to these conditions are marked seasonal fluctuations of $Ca^{2+}$ (monitoring well H31/41), $Cl^-$ (H52/53), $K^+$ (H41), $Mg^{2+}$ (H41), $Na^+$ (H31/41), $Si^{4+}$ (H41) and $SO_4^{2-}$ (H31/41).*

**And a discussion of the effect of contamination/pollution/human factors to data is desired.**

Hainich CZE is a rare non-contaminated region in Central Europe without villages, waste disposal sites or industry within the groundwater recharge area. Like vulnerability studies (investigating the impact of potential anthropogenic hazards), we used comparable tools/methods for exploring soil hydraulic properties, aquifer configuration and land management. In contrast to "anthropogenic hazard studies", our focus lies on the reconstruction of the subsurface structure by utilizing pristine groundwater quality data.

Influences of land management on groundwater quality is now described in a separate chapter (5.2.3), that now reads as follows (page 14, line 19 ff.):

*5.2.3 Influences of land management in the recharge area on groundwater quality*
*The forest areas as a source of groundwater from catchment-near summit/shoulder wells are confirmed by their aquifer outcrop areas (moTK-1, moM-1) within the managed (and partly unmanaged) forest. In contradiction, we detected potentially agriculture-related substances (NO₃⁻, K⁺, Cl⁻) in the wells drilled to all aquifer storeys in midslope location wells (H31/41), although these groundwater types are recharged mainly within the forest (up to 96.5 % forest). These surface signals are interpreted to be related to the cropland and village areas within the preferential recharge zones with sinkhole lineaments. Anoxic groundwater in aquifers (moM8/9) partially recharged from outcrop areas with agricultural and village land use, are likely attributed to microbial oxygen depletion resulting from the degradation of organic carbon or oxidation*

*of inorganic electron donors. For shallow wells in valleys (H42/43/4S), lateral soil water inflow with high organic/fertilizer load towards the valley provide an alternative way for the enhancement of oxygen depletion. Much lower $NO_3^-$, $K^+$, $Cl^-$ concentrations in the footslope wells (H52/53) are likely a consequence of the argillaceous caprocks that increase in thickness towards the footslope.*

Moreover we added a new chapter that deals with the intrinsic vulnerability of the recharge area (5.4). It reads as follows (page 17, line 19 ff.):

*5.4 Assessment of vulnerability factors*
*The configuration of the aquifer system in the hillslope setting also controls the groundwater resource vulnerability. With our multi-method investigation of the different subsurface compartments, we also revealed factors of the areas´ intrinsic vulnerability. Characteristics of intrinsic vulnerability are solely controlled by hydrogeological properties of the aquifer and overburden (Vrba and Zoporozec, 1994), and integrate the inaccessibility (i.e. by low-permeable cover strata) of the saturated zone and the attenuation capacity (retention, turnover) of the overburden (Adams and Foster, 1992).*
*The stacking of aquifers/aquitards, the coverage with caprocks and the lateral continuity of strata reduces the overall dominating disperse infiltration and, thus the intrinsic vulnerability. Threatening of single aquifer storeys or assemblages is predominantly controlled by fractures/faults (valleys) or karst phenomena bypassing the protective cover of soils and unsaturated zones (Fig. 8). A moderate degree of physical filtering by the narrow fractures, which are the predominant flow paths, is assumed. Also the claystone and marlstone interlayers bear a certain filtering of contaminants by retention. The preferential recharge/outcrop zones of the aquifer storeys are characterized by generally highest vulnerability. However, these zones, located in the summit to upper midslope of low mountain ranges, are mostly covered by forest. Generally, the summit position of outcrop zones of the main aquifer storey (moTK-1) lowers the risk for contamination, paradoxically, due to the thin soils that prevented lasting agricultural use and settlements. Further zones of higher vulnerability are located in the discharge of sinkhole lineaments or fracture zones that bypass surface water or drains directly into the aquifers. A mapping and structural investigation of these geostructural links will be recommended for (i.e.) proper dimensioning of drinking water protection zones. In comparison to classical karst sites (i.e. massive carbonates) with pronounced karst phenomena, (Doerfliger et al.,1999, Witowski et al., 2002), our portrayed setting shows an overall low to moderate vulnerability, that is locally elevated by inherent factors (outcrop zones) and inherited features (i.e. karstification of the underlying Middle Muschelkalk subgroup).*

**6) In the end of section 4.2, the authors classify three modes of subsurface water flow in the karst aquifer. I would say the lineaments of sinkholes are not necessarily due to flow through open faults. Is there a possibility that bedding parallel in either unconfined and confined aquifer can cause lineaments of sinkholes as well? Probably just track the faults/fracture zones from geological map/structure survey.**
We agree with the reviewer that a proper definition of caprock sinkholes is necessary for the interpretation of fault related flow. The paragraphs concerning caprock sinkholes now read as follows:

*Second order relief elements are also the two NW-SE oriented lineaments of more than eighty caprock sinkholes (which are passive karst phenomena) with up to 80 m in diameter. Caprock sinkholes are mostly exhibited on local ridges. A second and parallel lineament of three shallow elongated (uvala-like) karst depressions with a horizontal extent of up to 400 m crosses the lower Hainich hillslope* (page 7, line 30 ff.)

*The second route for preferential recharge is related to the lines of caprock sinkhole lineaments, which can be tracked over more than four kilometers in the midslope between transect locations H2/3 and H4. The origin of these sinkholes does not lie in the karstification of the Upper Muschelkalk strata itself, as the combination of aquifer fracture networks (tight conduits) and stabilizing, insoluble aquitard beds do not cause sufficient mass deficits that will allow hanging wall collapses. Here, caprock sinkholes are related to mass deficits by subrosion in the underlying evaporite rocks with gypsum and halite (Mempel, 1939; Malcher, 2014). As caprock sinkholes are arranged in lineaments (Fig. 2 A) parallel to regionally known fault orientations, it is very likely that they are coupled to the penetration of surface/subsurface water at fracture zones (compare to: Smart and Hobbs, 1986; Worthington, 1999; Klimchouk, 2005), that promote preferential recharge (Smart and Hobbs, 1986; Suschka, 2007). Although dissolution of soluble rocks is limited within the target aquifers of this study, it is reasonable, that collapse structures are accompanied by enhanced rock fracturing and permeability* (page 13, line 15 ff.)

*As the aquitard interbeds highly reduces vertical flow connections, outcrop zones of the aquifer storeys as well as caprock sinkholes act as important preferential infiltration pathways which are typical for hillslope recharge zones. Since caprock sinkholes remain dry, even directly after precipitation events, high infiltration rates can be assumed for these structures* (page 15, line 8 ff.).

We also tracked the lines of sinkholes and displayed these lineaments in Figure 2A (page 31). These zones are also included into the input dataset of the recharge potential map (Fig. 8 page 37). Mapped caprock sinkholes and uvala-like depressions are orientated in the same direction of NW-NE striking faults and to the geological fold axis (Hainich ridge, respectively), we interpret these linear features as fracture zones.

**7) Discussion 4.3 has weak relevant to the statistics analysis result. I don't think the authors have enough data to discuss karstification dissolution, so I recommend removing it.**
We agree with the reviewer and removed the chapter.

**8) To be honest, I didn't get the key points in the conclusion part. The authors do not need to mention the results of mapping and survey in the conclusion part. I suggest the authors to summarize the results of data analysis and emphasize the relationships. It might be better to make the statements by bullet points.**
We agree with the reviewer. The new paragraph is listed by bullet points and reads as follows (page 18, line 6 ff.):

- *Low-permeable marlstone beds within a marine succession of high lateral continuity represent a number of aquitards that cause a multi-storey hydrostratigraphy of the Upper Muschelkalk formations.*
- *As a multi-storey hydrostratigraphy exhibits limited vertical percolation, the outcrop zones of dipping aquifer storeys become very important as preferential surface-recharge areas for inputs of matter and energy.*
- *Diffuse fracture flow dominates over karst/conduit flow in the mixed/multi-layered lithology. Subsurface water flow predominantly takes place in bedding-plane parallel mode / in stratabound fractures of the limestone beds and it is trackable from the recharge areas along the storeys.*
- *From summit to footslope positions, travel distances and presumably groundwater ages generally increase. For the individual storeys however, travel distances to monitoring wells decrease in the downslope direction, whereas their groundwater ages very likely increase due to lower fracturing and higher retention. In the same direction, surface controls (i.e. nutrient input) decrease and subsurface controls (water-rock-interaction) increase.*
- *Compared to more vulnerable settings (i.e. massive carbonate karst, open karst), the mixed carbonate-siliciclastic alternations exhibit moderate intrinsic vulnerability. This is due to lateral continuity of low permeable interbeds,*

*soil covers and caprocks, of which the latter successively increase in thickness towards the footslope. Areas downstream the caprock sinkhole lineaments (and likely transverse valleys) are likely more threatened by anthropogenic (mostly agricultural) input.*

- *The quality of groundwater resources with peripheral hillslope recharge benefits from extensive land management or, ideally (managed/unmanaged) forest coverage and reveals the importance of recharge area protection.*

**2. Point-by-point response to review 2 (hess-2016-374-RC3)**

**The study is based on a very sound data set, the manuscript reads well and the joint discussion of the different data and their synthesis to conceptual model is appealing. But as it stands it remains a very sound and thorough study of a single case, because the authors miss quite obvious opportunities to address more generic questions.**

The study is on the one hand a comprehensive characterization of the subsurface structure of the Hainich CZE and also includes many general aspects which are transferable to other aquifer systems. We are sorry that the reviewer got the impression of missing generic questions. Thus we highlighted the thin-bedded mixed carbonate-siliciclastic rocks forming thin aquifer storeys. This important and very frequent hydrogeological setting of carbonate-/siliciclastic-rock alternations in hillslope terrain is no described in depth in the literature.

We added factors of intrinsic vulnerability (chapter 5.4, page 17, line 19 ff.) that reads as follows:

*5.4 Assessment of vulnerability factors*

*The configuration of the aquifer system in the hillslope setting also controls the groundwater resource vulnerability. With our multi-method investigation of the different subsurface compartments, we also revealed factors of the areas´ intrinsic vulnerability. Characteristics of intrinsic vulnerability are solely controlled by hydrogeological properties of the aquifer and overburden (Vrba and Zoporozec, 1994), and integrate the inaccessibility (i.e. by low-permeable cover strata) of the saturated zone and the attenuation capacity (retention, turnover) of the overburden (Adams and Foster, 1992).*

*The stacking of aquifers/aquitards, the coverage with caprocks and the lateral continuity of strata reduces the overall dominating disperse infiltration and, thus the intrinsic vulnerability. Threatening of single aquifer storeys or assemblages is predominantly controlled by fractures/faults (valleys) or karst phenomena bypassing the protective cover of soils and unsaturated zones (Fig. 8). A moderate degree of physical filtering by the narrow fractures, which are the predominant flow paths, is assumed. Also the claystone and marlstone interlayers bear a certain filtering of contaminants by retention. The preferential recharge/outcrop zones of the aquifer storeys are characterized by generally highest vulnerability. However, these zones, located in the summit to upper midslope of low mountain ranges, are mostly covered by forest. Generally, the summit position of outcrop zones of the main aquifer storey (moTK-1) lowers the risk for contamination, paradoxically, due to the thin soils that prevented lasting agricultural use and settlements. Further zones of higher vulnerability are located in the discharge of sinkhole lineaments or fracture zones that bypass surface water or drains directly into the aquifers. A mapping and structural investigation of these geostructural links will be recommended for (i.e.) proper dimensioning of drinking water protection zones. In comparison to classical karst sites (i.e. massive carbonates) with pronounced karst phenomena, (Doerfliger et al.,1999, Witowski et al., 2002), our portrayed setting shows an overall low to moderate vulnerability, that is locally elevated by inherent factors (outcrop zones) and inherited features (i.e. karstification of the underlying Middle Muschelkalk subgroup).*

We also compiled a recharge potential map (Fig. 8, page 37), that comprehensively displays recharge options for selected aquifer storeys.

Furthermore, we added general aspects of this aquifer type to the conclusions (page 18, line 6 ff.) that now reads:

- *Low-permeable marlstone beds within a marine succession of high lateral continuity represent a number of aquitards that cause a multi-storey hydrostratigraphy of the Upper Muschelkalk formations.*

- *As a multi-storey hydrostratigraphy exhibits limited vertical percolation, the outcrop zones of dipping aquifer storeys become very important as preferential surface-recharge areas for inputs of matter and energy.*

- *Diffuse fracture flow dominates over karst/conduit flow in the mixed/multi-layered lithology. Subsurface water flow predominantly takes place in bedding-plane parallel mode / in stratabound fractures of the limestone beds and it is trackable from the recharge areas along the storeys.*

- *From summit to footslope positions, travel distances and presumably groundwater ages generally increase. For the individual storeys however, travel distances to monitoring wells decrease in the downslope direction, whereas their groundwater ages very likely increase due to lower fracturing and higher retention. In the same direction, surface controls (i.e. nutrient input) decrease and subsurface controls (water-rock-interaction) increase.*

- *Compared to more vulnerable settings (i.e. massive carbonate karst, open karst), the mixed carbonate-siliciclastic alternations exhibit moderate intrinsic vulnerability. This is due to lateral continuity of low permeable interbeds, soil covers and caprocks, of which the latter successively increase in thickness towards the footslope. Areas downstream the caprock sinkhole lineaments (and likely transverse valleys) are likely more threatened by anthropogenic (mostly agricultural) inputs.*

- *The quality of groundwater resources with peripheral hillslope recharge benefits from extensive land management or, ideally (managed/unmanaged) forest coverage and reveals the importance of recharge area protection.*

*In general, for mixed carbonate-/siliciclastic rock alternations that are prone to develop multilayered aquifer systems, both the aquifer configuration (spatial arrangement of strata, hillside cutting, outcrop positions) and the related geostructural links (preferential recharge areas, karst phenomena) are major controls of impacting surface and subsurface factors. For the studied type of thin-bedded carbonate aquifer setting, we were able to demonstrate, that a comprehensive investigation of aquifer connectivity in the transit/discharge area as well as soil cover and land use in the recharge area is mandatory and must be rated indispensable for a thorough understanding of the state and evolution of groundwater quality* (page 18, line 25 ff.).

**Furthermore, the characterization appears not so holistic. With respect to the treatment of the soil, the assessment steps barely beyond a soil standard survey.**

In comparison to published examples, our study focuses on the assessment of groundwater quality by interpreting aquifer outcrop areas, aquifer stratigraphy, soil texture/thickness.

To increase the perceptibility of our holistic approach, we adapted our manuscript as follows:

- A soil map compiled on the basis of relief type, slope angle, surface geology and land use. Calibration of soil groups and soil texture/soil thickness was carried out by using our soil survey data (Figure 2B, page 31).

- Soil hydraulic conductivities (Ks) and grain size analysis measured on all relief positions and soil groups were be added to the dataset. The data is presented in Figure 3 (page 32). Spatial distributions of soil Ks are shown in Figure 8 (page 37). The additional paragraph concerning soil-Ks now reads as follows (page 8, line 30 ff.):

[revised manuscript text omitted]

**I thus encourage the authors to extend their analysis by addressing more generic questions using the beautiful data they have at hand. I hope the authors will find the following points helpful to further optimize their study for the reader and to fully explore the potential of their effort.**

**An obvious question that could be addressed is how much of the proposed experimental and monitoring effort is needed to come up with such a comprehensive assessment or how much of the information can be left out before without changing the quality of the conceptual model. This question could be easily addressed by taking the presented insights as the best guess of the unknown truth and stepwise leaving out increasing amounts of for instance their hydro chemical data (in space and or in time) and perform the same multivariate analysis.**

We agree with the editor, that this issue is necessary for the reader. Thus, we touched this issue in the conclusions. However, a full analysis of this question is a topic of ongoing research within the D03 project of CRC AquaDiva. The additional paragraph in the conclusions now reads (page 18, line 27 ff.):

*For the studied type of thin-bedded carbonate aquifer setting, we could to demonstrate, that a comprehensive investigation of aquifer connectivity in the transit/discharge area as well as soil cover and land use in the recharge area is mandatory and must be rated indispensable for a thorough understanding of the state and evolution of groundwater quality. Linking groundwater hydrochemistry mostly to surface factors such as land use would result in a contradiction, as groundwater chemistry does not reflect the type of land use in immediate proximity to the wells. Furthermore, footslope wells´ hydrochemistry is strongly impacted by aquifer stratigraphy, karst phenomena input and the cross-formational ascent of sulphatic groundwater that could not be evaluated without spatial lithostratigraphical data. A geostructural investigation and mapping is essential for the localization of aquifer outcrop areas and the assumption of stratigraphy-controlled flow directions. In case the soils group and the type of land use would have been neglected, discrimination between influences by natural and anthropogenic controls would have been ambiguous. If the dataset included hydrochemistry and multivariate statistics only, the interpretation of this dataset would have been ambiguous, also as different chemical surface/subsurface sources result in similar hydrochemical compositions.*

*Recent studies of the CRC AquaDiva focus on signal transit and transformations of surface signals. This is for instance applied to surface-sourced organic matter and microorganisms (Küsel et al., 2016, Schwab et al., 2017, Lazar et al., 2017) to further investigate subsurface connectivity, surface-subsurface interactions and functions of microbial life in groundwater environments. Further CZ exploration should also aim the investigation of deeper strata connection by regional groundwater flow, hydraulic properties and proportions of unsaturated zones and matter processing within.*

**The characterization of the soil is compared to the characterization of the deeper subsurface rather descriptive and follows mainly standard mapping approaches. I wonder whether any data on permeability and infiltrability where**

**collected?**

Among others, soil thickness data and soil hydraulic conductivities (duplicate measurements from 16 sites, 2 depths) were added to the dataset. The new paragraph reads as follows (page 8 line 30):

*4.2.2 Soil hydraulic properties*

*Average (Median) soil hydraulic conductivities (Ks) of the five major soil groups infer, that Rendzic Leptsols and Chromic Cambisols (2.5 to 5\* $10^{-5}$ m/s) form the most conductive soil cover in the study area, followed by Cambisols, Luvisols (1.3 to 1.5\* $10^{-4}$ m/s) and Stagnosols (about 6\* $10^{-7}$ m/s; Table 1, Fig. 3 and 8). Topsoils of Chromic Cambisols are considerably more conductive than those in Rendzic Leptosols and Luvisols. Generally subsoils are less conductive than topsoils of the same location. Soil hydraulic conductivities (Ks) are essentially uncorrelated with the soil texture (i.e. correlation Ks vs. Median grain size: Spearman $r^2 = + 0.17$). Soil thickness is uncorrelated to the slope gradient ($r^2 = -0,24$ and barely better if slope positions are correlated individually).*

Furthermore, the chapter on soils in the results section was extended. This paragraph now reads (page 8, line 13 ff.):

*4.2.1 Soil distribution and soil development*

*Soils cover the entire landscape with major soil series developed from carbonate rocks ("carbonate soil series": Rendzic Leptosols to Chromic Cambisols), siliceous rocks ("siliceous soil series": Luvisols, Stagnosols) or alluvial sediments (WRB and German soil groups: Table 1). Culmination areas and adjacent shoulder positions are covered with Cambisols. Small plateaus and spurs between transverse valleys in shoulder positions exhibit Chromic Cambisols which grade into Cambic Regosols on the shoulder and Calcaric Regosols on the midslope. Chromic Cambisols are also found in local depressions and old caprock sinkholes. Rendzic Leptosols occur in form of narrow patches in western crestal areas (close to well H11). Luvisols are coupled to the spatial distribution of loess loam in the central and eastern midslope/footslope areas. In case of a very thin loess loam cover, soils are developed as Pelosol-Cambisol and Cambisol. Fluvial soils cover the central parts of headwater areas and the complete valley floor in the lower parts (in the northeast) of the study area. Colluvisols occur at the margins of local valley flanks in the shoulder and midslope area (Fig. 2 B). Typical sequences of two superimposed soils comprise (I) Chromic Cambisols (paleosols) developed from marlstones and (II) Luvisols developed from loess loam (with windblown loess sedimentation after the formation of soil (I) (Fier, 2012). Average soil thicknesses are 21 cm (for Rendzic Leptosols), 32 cm (Rendzic Leptosol-Cambisol transitions), 66 cm (Cambisols), 54 cm (Chromic Cambisols), 71 cm (Luvisols), 132 cm (Fluvisols), 91 cm (Stagnosols) and 78 cm (Colluvisols). Subsoils show higher average clay and fine silt content compared to the topsoils of the same soil group (Fig. 3).*

**Even if, not the analysis of the available soil types lacks behind its potential to infer on recharge areas. The latter depend on the ks, retention properties and apparent preferential pathways and the spatial pattern thereof. Even if the latter were not mapped, one could use a pedo transfer function to estimate ks and compile a geostatistical analysis with interpolation or even conditional simulation. This would yield an estimate on potential hot spots for recharge.** We carried out soil hydraulic conductivity measurements as well as a grain size analysis for all soil groups and all types of land management. Correlations between soil hydraulic conductivities and mapped quantitative soil texture properties (i.e. median grain size, clay content, soil grainsize class) was not possible, as the soil hydraulic conductivities are uncorrelated to the soil texture in our study area. For this reason, a quantitative map of hydraulic conductivities (Ks) was not possible with the actual dataset. Instead of a Ks-map, we decided to display a recharge potential map (chapter 5.2.4) that considers the following parameters, which is described in the following section (page 7, line 14 ff):

*3.9 Groundwater recharge potential map*

*For the determination of preferential zones for infiltration/recharge, we calculated qualitative maps of the recharge potential (compare to Muir and Johnson 1979; Shaban et al., 2006; Deepa et al., 2016) in ArcGIS 10.3 by a weighted linear combination of surface/subsurface properties that influence water infiltration and percolation. The input raster datasets and the respective weight factors were chosen based on expert knowledge and the best fit to measured soil hydraulic conductivities: aquifer storey overburden thickness (40 %), type of bedrock (limestone-dominated vs. marlstone-dominated strata; 15 %), soil classes (15 %), fracture/karst zones (10 %); vegetation and type of land management (10 %), soil thickness (5 %) and slope angle classes (5 %).*

**Technical details:**

**- Generally the figure caption are very brief.**

We added more detail to all figure captions. The new figure captions read as follows (page 28 line 2 ff.):

[revised manuscript text omitted]

**Figure 4: how much variance is explained by the first two principle components? I didn't find it in the text, would be nice to provide it here.**

The information has been added (page 10, line 42 ff.):

*According to the PCA of the complete parameter set, the first two components (PC1 plus PC2) explain 54.9 % (variances of PC1: 34.6 % and PC2: 20.4 %), and according to the parameter set without redox sensitive parameters PC1 plus PC2 explain 62.1 % (variances of PC1: 35.4 % and PC2: 26.8 %) of the total variability, respectively.*

**- I guess the color code in Figure 5 represent the cluster in Figure 4. Would be helpful to add that to the caption, also of Figure 6.**

This was done accordingly (Figure 6 page 35, Figure 7 page 36, Figure 9 page 38).

**- Figure 7. Is the error bar the standard deviation of the sample or the root of the estimation variance of the average. I guess the latter is more appropriate to infer on significant differences.**

The error bar represented the standard deviation. Obviously we missed to mention this in the figure caption. After checking

both methods, we are confident, that the root of the estimation variance of the average is more appropriate. We changed the graph and figure caption, accordingly (Figure 9 page 38)

**- The reference in the section 4.1 (infiltration properties) is not the most recent one, particularly not with respect to preferential flow. Furthermore, this part is rather descriptive and could possibly be written without the data you have.**

We agree and changed the paragraph as suggested. The paragraph now reads (page 13, line 27 ff).:

*The spatial distribution of the soil groups reflect the outcrop zones of limestones/marlstones (Greitzke and Fiedler, 1996; Brandtner, 1997; Rau and Unger, 1997) and the succession of aquifer storeys in the summit/shoulder area of the hillslope which exhibits little quarternary rocks coverage and thin soils (Rendzic Leptosol, Chromic Cambisol) with favorable infiltration properties in the aquifer outcrop areas of moTK-1 and moM-1/6/8. Thicker relictic Chromic Cambisols which are formed by intensive decarbonatization (Rau and Unger, 1997; AG Boden, 2005) are restricted to accumulations in former depressions (i.e. caprock sinkholes) on shoulder regions of the hillslope (moTK-1 and moM-1 outcrops). Besides these depressions, Chromic Cambisols with low hydraulic conductivities are typically the subsoil layer of sequences with two layer superimposed soils, resulting in lateral soil interflow (Ali et al., 2011) as it is observed in the shoulder and midslope region during the measurement of soil hydraulic conductivities. Soils on unfractured marlstone/claystone aquitards are typically Calcaric Regosols and Stagnosols with low hydraulic conductivities of both, soils and parent rocks. According to our dataset, soil thickness is primarily controlled by the slope gradient of transverse valleys with increasing soil thickness and water storage options towards the center of valleys. As a result of the increasing coverage of parent rocks by loess loam from the regional midslope to the footslope, (related to solifluction of windblown dust; Kleber, 1991; Bullmann, 2010), loess loam becomes the dominating soil substratum in the outcrop areas of aquifer storeys moM 7-8-9. In this area, Luvisols cover local ridges and Luvisol-Stagnosol-transitions (related to the continuous vertical clay relocation (Rau and Unger, 1997) cover local valleys, whereas older caprock sinkholes are either filled by colluvisols or with loess loam, leading to either poor or very good infiltration/recharge properties. In the same context, Bachmair et al. (2009) considers the role of micro-depressions in soils as one of the major factors for preferential recharge.*

More actual literature has been added. The revised paragraph now reads as follows (page 12, line 3 ff):

*Although median grain size, grain size sorting and the standard devitation of grain size fractions is comparable in all soils, "carbonate series soils" (Rendzic Leptosol and Chromic Cambisol) offer higher hydraulic conductivities than "siliceous series soils" (Cambisol, Luvisol, Stagnosol) by one order of magnitude. Soil hydraulic conductivities (Ks) are uncorrelated to soil texture properties. For this reason a linear transfer function (Ks vs. median grain size or grain size category) for extending the 16 Ks measurements to the 271 mapped soil profiles was not possible with the actual dataset. A low correlation between Ks and texture is related to strong influences of soil structure and aggregation (Totsche et al., 2017) rather than texture on Ks. Structural parameters are for instance the content and proportion of macro and microaggregates, a hierarchical aggregate system, the presence and frequency of secondary pores and anthropogenic changes like former plough traces or traces of forestry machines. Bachmair et al. (2009) for instance identify tillage-related macropores as a main factor for deep infiltration into cropland soils, whereas surface littler layers inhibit infiltration in forest soils. Moreover preferential infiltration through biomacropores (i.e. earthworm borrows, root channels), that bypass the soil matrix are of great significance in aggregated and argillaceous soils with low matrix conductivity for the infiltration during heavy rain events (Deurer et al., 2003; Weiler and Naef, 2003; Blouin et al., 2013). Klaus et al.,(2013) identify vertical*

*macropores of anecic earthworms as a major flow control in a hillslope tile drain system. Wienhöfer and Zehe (2014) additionally consider loose, litter-rich topsoils, lateral preferential pathways, preferential pathways at the soil-bedrock surface and bedrock topography.*

**3. List of all changes made in the manuscript (HESS-2016-374)**

- We changed the title of the manuscript by focusing on the scientific questions and the geologic situation with thin bedded mixed carbonate/siliciclastic rocks as it was proposed by reviewer 1.

- The abstract was reformulated as it was suggested by reviewer 1 by focusing on the novelties of the comprehensive approach as well as on the major research outcomes (interpretation of groundwater quality by means of hydrostratigraphy, preferential infiltration zones and land management as well as soil hydraulic properties in the groundwater catchment).

- The introduction now includes published examples for comparable comprehensive investigations and their application for vulnerability assessment / for understanding groundwater chemistry. The special geological characteristics of the study site were highlighted in the introduction as proposed by reviewer 1. Moreover, the uniqueness and the capacities of Hainich CZE were explained in the introduction.

- All study aims were reformulated (as suggested by reviewer 1) to highlight scientific questions that are to be explained by this study. All three study aims are now connected to each other (structure of the critical zone, groundwater quality controls, and spatial variations of surface impact).

- The site description was condensed into one chapter that is now located between the introduction and the methods chapters, as proposed by reviewer 2. The definition of groundwater assemblages, defined by former publications (HTU/HTL) are mentioned in the site description, only, and not in the results and discussion chapter.

- The methods chapter was re-organized by sub-chapters (3.1 to 3.8). Hydrogeochemical methods, which are regarded to be more relevant in this study (hydrogeochemistry, statistical analysis) are grouped at the beginning of the methods chapters. The methods chapter was complemented by the newly added methods (measurement of grain size, measurement of soil hydraulic conductivities, and construction of the conceptual soil map).

- General chapters concerning relief and land management were reduced in supplementary information. A short explanation of caprock sinkholes was added to the relief chapter (3.1). The results chapter was re-organized in surface properties (4.2) and subsurface properties (4.3). Soil distribution and soil development is now described in the results chapter in a briefer style. Soil hydraulic properties which are necessary for understanding recharge potential are added to the results (4.2.2) as well as the geological outcrop zones (4.2.3).

- As soil characterization is rather descriptive (as it is remarked by reviewer 2), grain size data of 16 soil sites (up to 6 samples/Profile) and soil hydraulic conductivities are measured at 16 sites (2 depths/profile) were added to the results (4.2.1 Soil distribution and soil development; 4.2.2 Soil hydraulic properties). Moreover, soil mapping was complemented by a conceptual soil map.

- The hydrostratigraphy chapter (3.2) was reorganized starting with the core log correlation and the ten newly defined aquifer storeys. Aquifer compartments of former publications are mentioned in a briefer way at the end.

- For a more holistic characterization of the groundwater catchment, we added grain size classes within the outcrop areas / capture zones of all aquifer zones and soil hydraulic conductivities to the results chapter (4.2.2, 4.3.3).

- The paradox situation of shallow, anoxic and deep, oxic aquifers was added to the results chapter (4.4.1) as well as the variances of principal components as requested by reviewer 2. A brief chapter concerning groundwater level and groundwater quality fluctuations was added to the results chapter (4.4.3) as requested by reviewer 1. This chapter is intentionally short; as these issues are discussed in detail within a second research article.

- The discussion chapter now starts with the aquifer structure interpreted by lithology (5.1.1), followed by the newly written chapter concerning structural and relief information evaluated by groundwater chemistry/hierarchical cluster analysis and principal component analysis (5.1.2).

- More generic questions like the controls on groundwater quality in groundwater catchments (5.3), influences of soil/soil hydraulic conductivity (5.2.2) and land management (5.2.3) on groundwater quality and a conceptualization in form of vulnerability classes was added to the manuscript.

- The discussion chapters are now focused on the discussion of statistical analysis of geochemical data as requested by reviewer 1. In addition to the enhancement of the existing chapter 5.3 (Controls on groundwater quality: interpretation of hierarchical clusters), we added a new chapter on hierarchical cluster interpretation (5.1.2 Structural and relief information evaluated by groundwater chemistry).

- Preferential recharge areas are interpreted in a new chapter (5.2.1) that explains the nature of caprock sinkholes and outcrop zone infiltration. Influences of soil properties (5.2.2) as well as types of land management (5.2.3) and their influence on infiltration are discussed in two chapters. References on preferential flow in chapter 5.2.2 (Influences of soils in the recharge area) are updated with more recent literature citations as requested by reviewer 2.

- Four modes of groundwater flow (5.2.5) are described in detail by interpreting groundwater chemistry and hydrostratigraphy. This includes the effect of fault zones on groundwater quality as well as some literature citation as it was desired by reviewer 1.

- The interpretation of groundwater provenance (5.3) by interpreting the hierarchical clusters of hydrogeochemistry was reorganized. The interpretation now compares more comprehensively (as requested by reviewer 2) surface (land management, soil hydraulic properties, soil chemistry) and subsurface influences (aquifer rock lithology and aquifer bulk rock mineralogy, length of flowpaths) with focus on the provenance of dissolved substances for interpreting hydrogeochemical patterns. Modes of groundwater flow are transferred to chapter 5.2.5.

- More general aspects of groundwater recharge and hydrostratigraphy in thin bedded carbonate-siliciclastic alternations were added to the conclusions. The additional chapter concerning vulnerability factors (5.4) as well as to the figure showing vulnerability classes were added the discussion. This chapter includes a brief discussion of intrinsic vulnerability as requested by reviewer 1. The associated recharge potential map contains the tracked faults and fracture zones from the observations of sinkhole lineaments and the geological map as proposed by reviewer 1.

- We removed the last chapter in the discussion, concerning the controls for variations in karstification (as proposed by reviewer 1) as this chapter is not directly related to the study aims.

- Conclusions were reformulated and now summarize the results of hydrostratigraphy, the interpretation of hierarchical clusters, the interpreted modes of groundwater flow and intrinsic vulnerability. Statements are listed with bullet points as proposed by reviewer 1.

- A short discussion of how much information/monitoring effort is necessary to come up with the actual conceptual model (as it is proposed by reviewer 2) is described at the end of the conclusions. This question is not discussed in detail, as it is a topic of ongoing research within the D03 project of AquaDiva.

- All figure captions are extended as it is proposed by reviewer 2. Also color codes and variances of principal components as well as estimated variances of the average are added to the figures/the text.

Figures:

- o The map concerning soil groups and land management types was complemented by a conceptual soil map, interpolation lines of soil thickness and soil hydraulic conductivities (Ks). For a greater lucidity, the map was split into two thematic maps.

- o Surface/subsurface properties of the aquifer outcrop zones were displayed in an additional graph showing the frequency of land management types, soil groups and grain size classes.

- o Physical soil properties (abundances of fine grain size fractions and Ks) were displayed in an additional graph for Topsoil/subsoil and bedrock material.

- o The groundwater classification (piper diagram) was removed, as the groundwater classification is fully described in the test.

- o A recharge potential map for the moTK-1 and moM-8 aquifer storey was added.
  The former map of outcrop zones of all aquifer storeys, surface karst phenomena, groundwater isolines and rivers was removed.

[revised manuscript text omitted]

**Copyright statement**

The authors grant, that they are authorized by their co-authors to enter into the following arrangements. The work described has not been published before (except in the form of an abstract or proceedings-type publication – including discussion papers – or as part of a published lecture or thesis). It is not under consideration for publication elsewhere, and its

20 publication has been approved by all the authors and by the responsible authorities – tacitly or explicitly – of the institutes where the work was carried out. The authors have secured the right to reproduce any material that has already been published or copyrighted elsewhere. The authors agree to the following license and copyright agreement:

The copyright of any article is retained by the authors. Authors grant Copernicus Publications a license to publish the article and identify itself as the original publisher. Authors grant Copernicus Publications commercial rights to produce hardcopy

25 volumes of the journal for purchase by libraries and individuals. Authors grant any third party the right to use the article freely under the stipulation that the original authors are given credit and the appropriate citation details are mentioned. The article is distributed under the Creative Commons Attribution 3.0 License. Unless otherwise stated, associated published material is distributed under the same license.

[revised manuscript text omitted]